# Neutrophils induce paracrine telomere dysfunction and senescence in ROS-dependent manner

Anthony Lagnado[1,2], Jack Leslie[3], Marie-Helene Ruchaud-Sparagano[4], Stella Victorelli[1,2], Petra Hirsova[5], Mikolaj Ogrodnik[1,2], Amy L Collins[3], Maria Grazia Vizioli[1,2], Leena Habiballa[1,2,6], Gabriele Saretzki[7], Shane A Evans[8], Hanna Salmonowicz[1,2,7], Adam Hruby[1,2], Daniel Geh[3], Kevin D Pavelko[9], David Dolan[10], Helen L Reeves[11], Sushma Grellscheid[10,12], Colin H Wilson[13], Sanjay Pandanaboyana[13], Madison Doolittle[2], Thomas von Zglinicki[7], Fiona Oakley[3], Suchira Gallage[14], Caroline L Wilson[3], Jodie Birch[7] (iD), Bernadette Carroll[15], James Chapman[7], Mathias Heikenwalder[14], Nicola Neretti[8], Sundeep Khosla[2], Claudio Akio Masuda[1,2,16], Tamar Tchkonia[2] (iD), James L Kirkland[2], Diana Jurk[1,2], Derek A Mann[3,*] (iD) & João F Passos[1,2,**] (iD)

## Abstract

Cellular senescence is characterized by an irreversible cell cycle arrest as well as a pro-inflammatory phenotype, thought to contribute to aging and age-related diseases. Neutrophils have essential roles in inflammatory responses; however, in certain contexts their abundance is associated with a number of age-related diseases, including liver disease. The relationship between neutrophils and cellular senescence is not well understood. Here, we show that telomeres in non-immune cells are highly susceptible to oxidative damage caused by neighboring neutrophils. Neutrophils cause telomere dysfunction both *in vitro* and *ex vivo* in a ROS-dependent manner. In a mouse model of acute liver injury, depletion of neutrophils reduces telomere dysfunction and senescence. Finally, we show that senescent cells mediate the recruitment of neutrophils to the aged liver and propose that this may be a mechanism by which senescence spreads to surrounding cells. Our results suggest that interventions that counteract neutrophil-induced senescence may be beneficial during aging and age-related disease.

**Keywords** aging; neutrophils; senescence; telomeres
**Subject Categories** Cell Cycle; Immunology; Molecular Biology of Disease
**The EMBO Journal (2021) 40: e106048**
See also: **MSJ Burbano et al** (May 2021)

## Introduction

Neutrophils play a critical role in the process of microbial killing, which they achieve through the actions of a variety of neutrophil-expressed microbicidal factors including proteolytic enzymes and reactive oxygen species (ROS). In addition, neutrophils are rapidly recruited to sites of tissue damage in response to damage-associated molecular patterns (DAMPs) and their induction of neutrophil chemoattractants which include C-X-C chemokines, calprotectins, and lipid mediators important in the sterile immune response to tissue damage (Peiseler & Kubes, 2019). In acute resolving injuries, neutrophils stimulate wound repair by removing injured cells,

1   Department of Physiology and Biomedical Engineering, Mayo Clinic, Rochester, MN, USA
2   Robert and Arlene Kogod Center on Aging, Mayo Clinic, Rochester, MN, USA
3   Newcastle Fibrosis Research Group, Biosciences Institute, Newcastle University, Newcastle upon Tyne, UK
4   Translational Research Institute, Newcastle University, Newcastle upon Tyne, UK
5   Division of Gastroenterology and Hepatology, Mayo Clinic, Rochester, MN, USA
6   NIHR Newcastle Biomedical Research Centre, Newcastle University and Newcastle Upon Tyne Hospitals NHS Foundation Trust, Newcastle upon Tyne, UK
7   Ageing Research Laboratories, Faculty of Medical Sciences, Biosciences Institute, Newcastle University, Newcastle upon Tyne, UK
8   Department of Molecular Biology, Cell Biology and Biochemistry, Brown University, Providence, RI, USA
9   Department of Immunology and Immune Monitoring Core, Mayo Clinic, Rochester, MN, USA
10  Computational Biology Unit, University of Bergen, Bergen, Norway
11  Newcastle University Translational and Clinical Research Institute, Liver Unit, Newcastle upon Tyne NHS Foundation Trust, Newcastle upon Tyne, UK
12  Department of Biosciences, Durham University, Durham, UK
13  Department of Hepatobiliary Surgery, Newcastle upon Tyne Hospitals NHS Foundation Trust, Newcastle upon Tyne, UK
14  Division of Chronic Inflammation and Cancer, German Cancer Research Center (DKFZ), Heidelberg, Germany
15  School of Biochemistry, University of Bristol, Bristol, UK
16  Instituto de Bioquímica Médica Leopoldo de Meis, Universidade Federal do Rio de Janeiro, Rio de Janeiro, Brazil
    *Corresponding author. Tel: +44 191 208351; E-mail: Derek.Mann@ncl.ac.uk
    **Corresponding author. Tel: +1 507 293 9785; E-mail: Passos.Joao@mayo.edu

clearing cellular remnants, and releasing growth factors and pro-angiogenic molecules. In this context, neutrophils are important for resolution of inflammation and enabling the subsequent regenerative response. Where damage is chronic, neutrophils fail to effectively resolve inflammation and can instead exacerbate cellular stress and injury through their release of matrix metalloproteinases and ROS. Less well understood is the degree to which neutrophilic inflammation may induce changes in tissues that impact on longer-term health and susceptibility to disease.

Cellular senescence refers to an irreversible cell cycle arrest, which can be triggered by a number of stressors, including oxidative stress and telomere dysfunction among many others (Gorgoulis et al, 2019). Senescence is associated with a number of distinctive phenotypes, including the development of a pro-inflammatory response, commonly known as the senescence-associated secretory phenotype (SASP). The SASP is thought to communicate with the immune system and facilitate immune surveillance, stimulating the clearance of senescent or pre-malignant cells (Ovadya et al, 2018). However, chronic exposure to the SASP can lead to age-associated tissue dysfunction (Jurk et al, 2014; Xu et al, 2018). Consistent with this idea, senescent cells have been shown to accumulate in many tissues with age and at etiological sites in multiple chronic diseases (Ogrodnik et al, 2019a). Importantly, clearance of senescent cells either genetically or by using "senolytic" drugs has been shown to alleviate the onset of a number of pathologies during aging and disease (Zhu et al, 2015; Baker et al, 2016; Farr et al, 2017; Ogrodnik et al, 2017; Bussian et al, 2018; Musi et al, 2018; Anderson et al, 2019; Ogrodnik et al, 2019b).

Senescent cells have been shown to accumulate during aging in murine liver (Wang et al, 2009; Hewitt et al, 2012) and in the context of age-related liver disease in humans (Ogrodnik et al, 2017). The pathophysiological impacts of senescence are exemplified by the aging liver where regeneration and capacity to overcome injuries wanes in older animals and humans. Consequently, aging is associated with a higher incidence of acute liver failure and chronic liver disease. Consistent with a role for senescent cells in liver pathology, it has been shown that clearance of senescent cells reduces liver steatosis in aged, obese, and diabetic mice (Ogrodnik et al, 2017). However, the underlying mechanisms driving senescence in the liver are not completely understood. Here, we investigated the hypothesis that neutrophils recruited from the circulation in response to damage can act as drivers of cellular senescence and as such contribute to organ dysfunction during aging and disease.

We first observed that short-term co-culture of neutrophils with human primary fibroblasts accelerates the rate of telomere shortening and induces premature replicative senescence in the latter, a process which is dependent on ROS and requires cell-to-cell contact. Importantly, we found that ectopic expression of the catalytic subunit of telomerase (hTERT) prevents neutrophil-induced telomere shortening and premature senescence. Ex vivo studies involving precision-cut liver slices (PCLS) obtained from human subjects confirm that hepatic infiltration by neutrophils induces telomere dysfunction and senescence markers in liver hepatocytes. Further investigation involving mouse PCLS showed that neutrophils from transgenic mice overexpressing human catalase in mitochondria (MCAT) did not induce paracrine telomere dysfunction or expression of p21 as opposed to wild-type neutrophils. In vivo, we found that depletion of neutrophils reduces telomere dysfunction and

senescence in a model of acute liver injury. Finally, we observed that during liver aging, neutrophils accumulate in the hepatic parenchyma and are found in close proximity to hepatocytes containing dysfunctional telomeres. Genetic clearance of p16$^{Ink4a}$-positive cells in aged INK-ATTAC mice reduces the number of neutrophil infiltrates. In contrast, stimulation of neutrophil recruitment to the liver enhances hepatocyte senescence. Altogether, our findings suggest that neutrophils are both recruited by senescent cells and can be drivers of senescence. Therapies which counteract neutrophil-induced senescence may be beneficial during aging and age-related disease.

## Results

### 1-Neutrophils induce paracrine senescence

In order to test the hypothesis that neutrophils contribute to cellular senescence, we began by co-culturing human MRC5 fibroblasts for 3 days with freshly isolated neutrophils from similarly aged healthy donors (Figs 1A, and EV1A and B). To avoid the confounding factor of neutrophil cell-death (which we observed to occur after 24 h), we replaced neutrophils daily and cultured them at 3% O$_2$. In order to test the role of ROS as a mediator of neutrophil-induced senescence, neutrophils and fibroblasts were co-cultured in the presence or absence of extracellular recombinant catalase for 3 days. After 3 days of co-culture, neutrophils were removed (as well as the catalase) and MRC5 fibroblasts were passaged until they reached replicative senescence (Fig 1A).

Growth curves revealed that co-culture with neutrophils did not acutely affect the proliferation rate of MRC5 fibroblasts; however, instead it led to a premature growth arrest at later stages in culture, which could be prevented by pre-incubation with catalase (Fig 1B and C). We also observed that when neutrophils were LPS-primed the effects on replicative lifespan were more pronounced (Figs 1C and EV1C), which may be due to neutrophil activation and an associated stimulation of ROS (Fig EV1D).

Analysis of the proliferation marker Ki67 after 28 days in culture confirmed that fibroblasts co-cultured with neutrophils displayed reduced proliferation (Fig 1D and E). In contrast, senescent markers p16$^{INK4A}$ (Fig 1D and F) and SA-β-Gal (Fig 1D and G) were elevated in co-cultured fibroblasts at 28 days and these effects were prevented by treatment with catalase. p21 expression showed a trend for an increase with activated neutrophils and reduction with catalase treatment (Fig 1H). We observed in one experiment that neutrophil co-culture led to an increase in p15 expression at the 28-day time point and a reduction with catalase (Fig EV1E) but this finding was not reproducible in independent experiments (Fig EV1F).

Enhanced mitochondrial ROS generation is both a characteristic of senescent cells (Correia-Melo et al, 2016), but also can be a driver of senescence (Passos et al, 2007). We observed that fibroblasts that had been co-cultured with neutrophils displayed increased MitoSOX fluorescence, which was significantly reduced in catalase-treated co-cultures (Fig 1I). An important phenotype of senescent cells is the development of the SASP. In order to characterize the SASP, we carried out a cytokine array in conditioned media (CM) collected from these cells after 8 and 28 days of culture. From the panel of 48

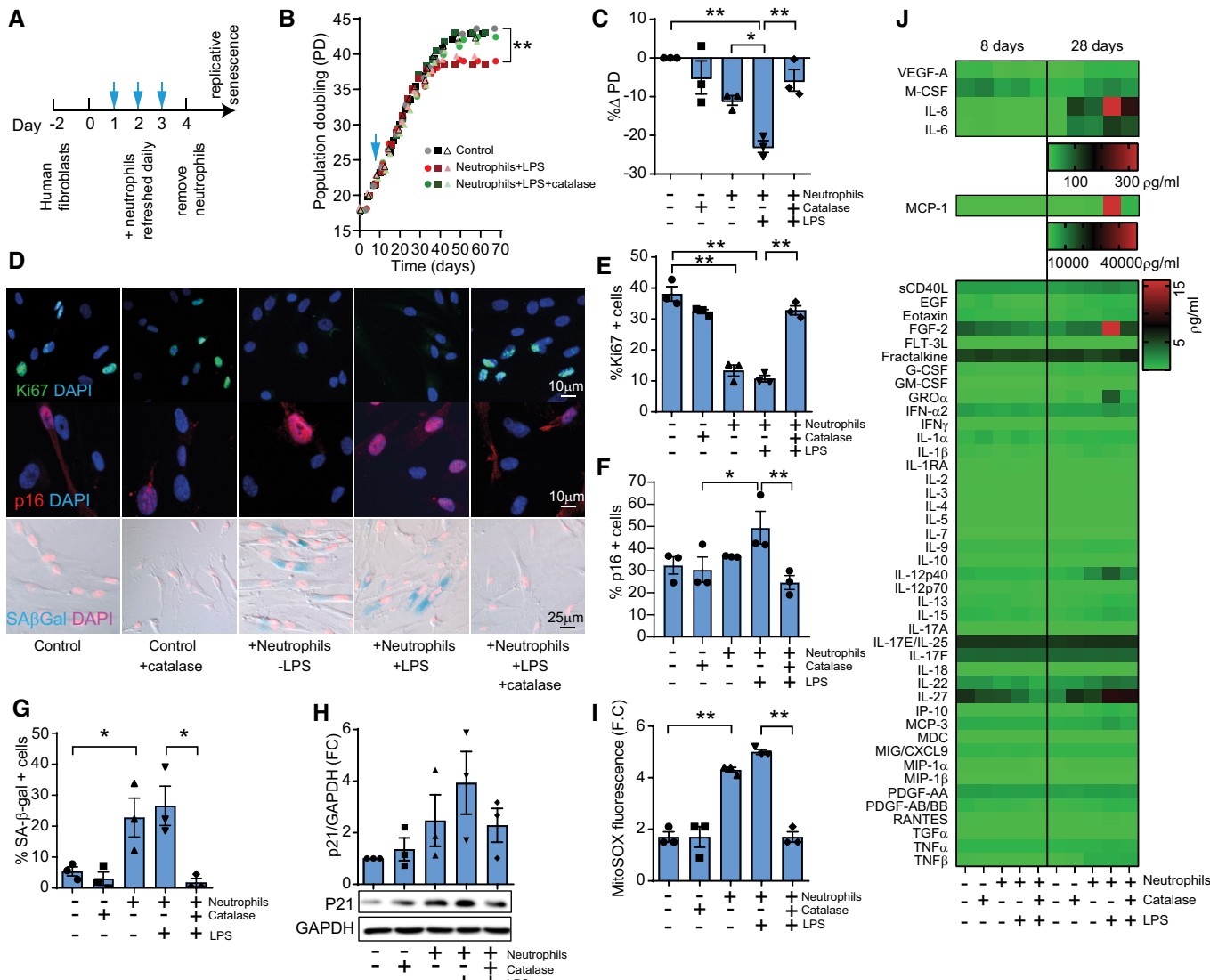

**Figure 1. Neutrophils induce paracrine senescence.**

A  Schematic depicting experimental design.
B  Effect of neutrophils and recombinant catalase on population doublings (data are from three independent cultures are shown). Blue arrow indicates time point in which neutrophils were added to culture for consecutive 3 days.
C  % Δ Population Doublings (PD) following neutrophil co-culture.
D  Representative images of Ki67 and p16[INK4A] immunofluorescence and SA-β-Gal staining.
E  % of Ki67-positive cells (28 days after co-culture).
F  % of p16[INK4A]-positive cells (28 days after co-culture).
G  % of SA-β-Gal-positive cells (28 days after co-culture).
H  Western blotting for p21 and quantification.
I  MitoSOX fluorescence.
J  Heat map depicting expression of 48 human cytokines in conditioned media at 8 and 28 days following co-culture.

Data information: Data are mean ± SEM of three independent experiments. Statistical analysis was performed using one-way ANOVA (Holm–Sidak method) for multiple comparisons and two-tailed *t*-test for single comparisons. **$P < 0.01$; *$P < 0.04$.
Source data are available online for this figure.

soluble proteins analyzed, we found that several factors were elevated in human fibroblasts that had been co-cultured with activated neutrophils and that catalase was able to prevent this effect (Figs 1J and EV1G).

Altogether, our results suggest that exposure of a bystander cell to neutrophils, while being insufficient to acutely affect proliferation, can limit its replicative lifespan and that this effect is predominantly mediated by ROS.

## 2- Neutrophils induce paracrine senescence *via* telomere dysfunction

Telomere shortening is believed to be a major contributor to replicative senescence (Harley *et al*, 1990; Bodnar *et al*, 1998). Moreover, previous work has shown that oxidative stress accelerates the rate of telomere shortening (von Zglinicki, 2002) and that telomeres are particularly sensitive to oxidative modifications (Petersen *et al*, 1998). In order to investigate whether neutrophils can induce telomere dysfunction in neighboring cells, we co-cultured neutrophils with human MRC5 fibroblasts as previously described and evaluated telomere length by Q-FISH in metaphase spreads at different time points. We found that neutrophils (both primed or non-primed) significantly reduced telomere FISH intensity over the course of 16 days, which is indicative of accelerated telomere shortening (Fig 2A–C). Consistent with a role for ROS in the process, 3-day pre-incubation with recombinant catalase was sufficient to prevent the accelerated telomere shortening.

Critically short telomeres induce senescence via the activation of a DNA damage response (DDR) (d'Adda di Fagagna *et al*, 2003). Furthermore, our previous work demonstrated that telomere DNA damage is irreparable and leads to a persistent DDR during cellular senescence (Fumagalli *et al*, 2012; Hewitt *et al*, 2012; Anderson *et al*, 2019). We therefore investigated the impact of neutrophils on telomere dysfunction in human fibroblasts by using Immuno-FISH to quantify co-localization between DDR proteins γH2A.X (or 53BP1) and telomeres, hereafter referred to as telomere-associated foci (TAF). We found that neutrophils induced a significant increase in TAF, which could be prevented by catalase (Fig 2D and E). It should be noted that we also found an increase in the frequency of total γH2A.X foci (Fig EV2A). Similarly, DNA breaks measured by COMET assay increased in fibroblasts that had been co-cultured with neutrophils, and the effect was rescued by catalase (Fig EV2B). However, suggesting a specific role for telomeres in the process, we found that MRC5 fibroblasts ectopically expressing the catalytic subunit of telomerase hTERT did not experience neutrophil-induced premature growth arrest (Fig 2F) or telomere shortening (Fig 2G). Additionally, we found that hTERT-expressing fibroblasts did not show increased TAF (Fig 2H and I), expression of senescent markers p21, IL-6, and IL-8 or decreased expression of Lamin B1 (Fig 2J–M), increased 53BP1 foci (Fig EV2C) or expression of p16 and p15 (Fig EV2D and E) 20 days after co-culture with neutrophils. Previous work had shown that under conditions of stress, telomerase can be translocated from the nucleus to the mitochondria improving mitochondrial function and reducing ROS through non-canonical roles (Ahmed *et al*, 2008). We did not find any evidence for translocation of telomerase to the cytosol as a result of neutrophil-induced stress (Fig EV2F).

Following the observation that neutrophils can induce paracrine senescence in fibroblasts, we sought to investigate if direct cell-to-cell contact or if soluble factors released by neutrophils are required for this effect. In order to explore these hypotheses, we co-cultured human fibroblasts with neutrophils for 3 days as described before and in parallel exposed human fibroblasts to conditioned media from neutrophils or co-cultured neutrophils and fibroblasts in separate layers sharing the same medium using transwell inserts (Fig EV3A). We observed that human fibroblasts that had been in direct contact with neutrophils for 3 days experienced a premature growth arrest starting at 20 days of culture, while fibroblasts treated with conditioned medium or in transwells continued proliferation (Fig EV3B). Furthermore, our results indicate that direct cell-to-cell contact between neutrophils and fibroblasts is necessary to induce TAF (Fig EV3C–E), p21 and p16 expression (Fig EV3F–I) and expression of SASP factors IL-6 and IL-8 (Fig EV3J and K).

In summary, our data indicate that ROS produced by neutrophils can accelerate the rate of telomere shortening in bystander cells, thereby inducing premature senescence and that this requires direct cell-to-cell contact.

## 3- Neutrophils induce telomere dysfunction and senescence in precision-cut liver slices

Precision-cut liver slices (PCLS) are useful tools to investigate liver function, since they retain the structure and cellular composition of the native liver thereby being largely superior to two-dimensional cultures. We have recently developed a bioreactor platform, which can maintain functional PCLS cultures for longer periods of time than other systems (up to 6 days). Using this methodology, we sought to investigate if neutrophils could induce telomere dysfunction and senescence in human livers.

In order to do this, we first isolated human liver tissue from the normal margins of colorectal metastasis resections from patients and generated PCLS. Slices were cultured in BioR plates in our patented bioreactor platform, as previously described (Paish *et al*, 2019). We then isolated human neutrophils from the peripheral blood of healthy donors and exposed PCLS to $1 \times 10^6$ neutrophils for 2 days (these were replenished every 24 h) (Fig 3A). We first confirmed that neutrophils infiltrated the 3D liver slice by immunohistochemistry against neutrophil marker CD66B (Fig 3B and C). We also monitored lactate dehydrogenase (LD) activity during the time of culture, which can be an indicator of cell-death. There was a tendency for reduction of LD activity in all conditions over the time course of the experiment, which may reflect an adaptation to culture conditions, but no significant change following neutrophil addition (Fig 3D). Similarly, TUNEL staining revealed that neutrophils did not induce cell-death in PCLS (Appendix Fig S1A).

We then performed 3D Immuno-FISH and Q-FISH to evaluate TAF and telomere length, respectively. We found that the addition of neutrophils to PCLS resulted in increases in the mean number of TAF and % of TAF-positive hepatocytes (Fig 3E–G). However, we failed to detect any effects on telomere length (Appendix Fig S1B). Finally, we found that exposure to neutrophils induced senescence markers p21 and p16^INK4A in PCLS (Fig 3H and I).

In order to further investigate the hypothesis that ROS derived from neutrophils contributes to bystander telomere dysfunction and senescence in liver, we isolated livers from 8- to 12-week-old male mice and generated PCLS as previously described (Paish *et al*, 2019). We then isolated neutrophils from the bone marrow of wild-type and *MCAT* transgenic mice and added them to the PCLS (Fig 4A). *MCAT* mice have a CMV enhancer/chicken beta-actin promoter driving the expression of human catalase gene in mitochondria and show reduced ROS in different tissues (Schriner *et al*, 2005). We first confirmed that neutrophils derived from *MCAT* mice express high levels of human catalase (hCAT) (Fig 4B), show increased catalase activity (Fig 4C), and show reduced DHR fluorescence following PMA stimulation (Fig 4D). After establishing that neutrophils from wild-type and *MCAT* mice infiltrated PCLS at

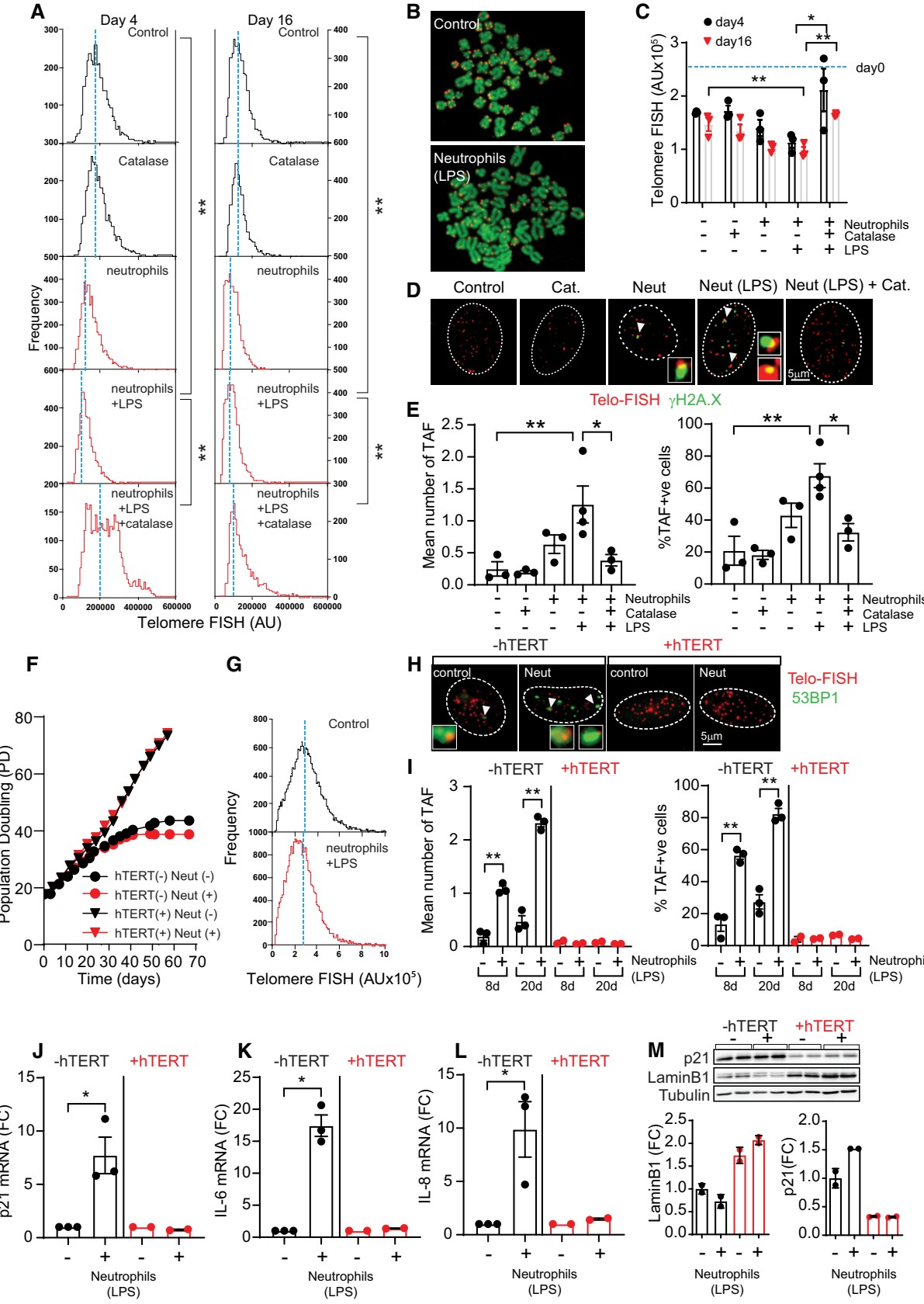

Figure 2.

**Figure 2. Neutrophils induce paracrine senescence via telomere dysfunction.**

A  Histograms representing distribution of individual telomere FISH intensities at day 4 and 16 following co-culture with neutrophils. Dotted blue lines represent median telomere FISH fluorescence. Data are from $n = 3$ independent experiments.

B  Representative micrographs of telomere FISH in metaphases.

C  Mean telomere FISH intensity at 4 and 16 days following co-culture with neutrophils represented as mean $\pm$ SEM of three independent experiments.

D  Representative Immuno-FISH micrographs (using telomere-specific (CCCTAA) peptide nucleic acid probe and anti-γH2A.X antibody). Arrows indicate co-localization between telomeres and γH2A.X.

E  Mean number and % of telomere-associated DDR foci (TAF). Data are mean $\pm$ SEM of three independent experiments.

F  Population doublings of hTERT-expressing MRC5 fibroblasts or controls following co-culture with primed neutrophils (data are representative of six independent experiments).

G  Histograms representing distribution of individual telomere FISH intensities in MRC5 fibroblasts expressing hTERT 16 days following neutrophil co-culture. Dotted blue lines represent median telomere FISH fluorescence.

H  Representative Immuno-FISH micrographs (using telomere-specific (CCCTAA) peptide nucleic acid probe and anti-53BP1 antibody) 20 days following co-culture.

I  Quantification of mean TAF and % TAF in MRC5 fibroblasts 8 and 20 days following neutrophil co-culture. Data are mean $\pm$ SEM of three independent experiments.

J–L  mRNA expression of p21 (J), IL-6 (K), and IL-8 (L) 20 days after neutrophil co-culture; data are mean $\pm$ SEM of 2–3 independent experiments.

M  Western blot and quantification of p21 and Lamin B1 20 days after neutrophil co-culture. Data are mean $\pm$ SEM of two independent experiments.

Data information: Statistical analysis was performed using one-way ANOVA (Holm–Sidak method) for multiple comparisons and Mann–Whitney test and two-tailed $t$-test for single comparisons. $**P < 0.01$; $*P < 0.04$.

Source data are available online for this figure.

similar frequencies (Fig 4E and F), we proceeded to investigate their role in induction of paracrine telomere dysfunction. We found that the addition of wild-type neutrophils to murine PCLS resulted in significant increases in the mean number of TAF and % of TAF-positive hepatocytes, but this effect was abrogated when MCAT-derived neutrophils were added (Fig 4G and H). We found no significant differences in total γH2A.X foci number or apoptosis (measured by LD activity) in any of the conditions (Fig 4I and J). Consistent with induction of senescence-associated pathways, we found a significant increase in the p21-positive cells following exposure to wild-type but not *MCAT*-derived neutrophils (Fig 4E and K).

Altogether, these data further support our *in vitro* findings that neutrophils can induce bystander telomere dysfunction and senescence markers in the liver in a ROS-dependent manner.

## 4- Neutrophils induce telomere dysfunction and senescence-associated phenotypes in a model of acute liver injury *in vivo*

In order to investigate the role of neutrophils in the induction of senescence *in vivo*, we used the well-established model of induction of acute liver injury and wound repair *via* the chemical carbon tetrachloride ($CCl_4$). Mice were pre-treated with neutrophil neutralizing antibody against Ly6G (or with anti-IgG control) for 12 h before $CCl_4$ injection (single intraperitoneal injection of $CCl_4$ at a dose of 2 μl/g body weight) and were humanely killed 8 h and 48 h after injection (Fig 5A). As shown previously (Moles *et al*, 2014), livers exhibited increased neutrophil infiltrates (measured by NIMP1) at 48 h, which were significantly reduced by anti-Ly6G (Fig 5B). We also found that hepatocytes positive for proliferation marker PCNA were increased 48 h after $CCl_4$ injection, indicating increased compensatory proliferation and remodeling. However, neutrophil depletion did not affect significantly the frequency of PCNA-positive hepatocytes (Fig 5C). Consistent with our previous data indicating that neutrophils can induce telomere dysfunction in neighboring cells, we found that the mean number of TAF and the % of hepatocytes positive for TAF increased after injury in IgG-treated mice but were significantly reduced in the mice pre-treated with anti-Ly6G (Fig 5D–F). We also evaluated the % of p21-positive hepatocytes and found that these increased

significantly at 48 h, but were reduced by anti-Ly6G (Fig 5D and G). Similar results were obtained when analyzing hepatocytes that were p21-positive but negative for the proliferation marker PCNA, which may be a more reliable indication of senescence (Lawless *et al*, 2010) (Fig 5H). RNA-ISH revealed that p16$^{Ink4a}$ mRNA levels were elevated 48 h after $CCl_4$ and anti-Ly6G-mediated neutrophil depletion reversed this increase (Fig 5D and I). mRNA expression of SASP factors Cxcl1 and IL-1α, detected by RNA-ISH, also showed a significant decrease at 48 h with neutrophil depletion (Figs 5J and EV4A). Loss of Lamin B1, another marker of senescence, did not show any significant difference in any of the groups (Fig EV4B and C).

Another feature of hepatocyte senescence is karyomegaly. We analyzed hepatocyte nuclear size by morphometric analysis of 4,6-diamidino-2-phenylindole (DAPI)-stained liver sections and quantified the % of karyomegalic hepatocytes, as before (Ogrodnik *et al*, 2017). Consistent with data for TAF, p16$^{Ink4a}$, and p21, we found an increase in the % of karyomegalic hepatocytes at 48 h and a significant reduction by neutrophil depletion (Fig 5K). Supporting the concept that senescent hepatocytes exhibit karyomegaly, we found that p21- and TAF-positive hepatocytes had higher nuclear area than negative ones (Fig EV4D and E). We next analyzed senescence-associated distension of satellites (SADS), an established marker of senescence and found that the mean number of SADS increased in hepatocytes 48 h after $CCl_4$ and was reduced with anti-Ly6G treatment (Fig 5L).

Our *in vitro* and *ex vivo* data suggest that oxidative stress is involved in neutrophil-induced telomere dysfunction and senescence; thus, we evaluated by 3D Immuno-FISH the presence of 8-oxo-2'-deoxyguanosine (8-oxodG) foci and its association with telomeres. Previous work has shown that telomeres are sensitive to ROS-mediated 8-oxodG formation that can contribute to telomere dysfunction (Fouquerel *et al*, 2019). We found that co-localization between 8-oxodG and telomeres was increased 48 h after $CCl_4$ and reduced after treatment with anti-Ly6G (Fig 5M and N) but total number of 8-oxodG foci was not changed (Appendix Fig S2). Finally, we evaluated telomere length (by Q-FISH) and found no statistically significant differences between groups (Fig 5O).

Previous work had demonstrated a specific role for Toll-like receptor 2 (TLR2) in the recruitment of neutrophils after CCl₄-induced liver injury (Moles *et al*, 2014). Accordingly, we also observed reduced TAF in hepatocytes from $Tlr2^{-/-}$ mice following CCl₄, but no changes in the frequency of proliferating hepatocytes

(Fig EV4F–K). TLR2 has recently been implicated in the regulation of senescence arrest and the SASP; thus, it is entirely possible that reduced TAF in hepatocytes is a direct consequence of deletion of TLR2 rather than an indirect effect of impaired neutrophil recruitment (Hari *et al*, 2019; Jin *et al*, 2020).

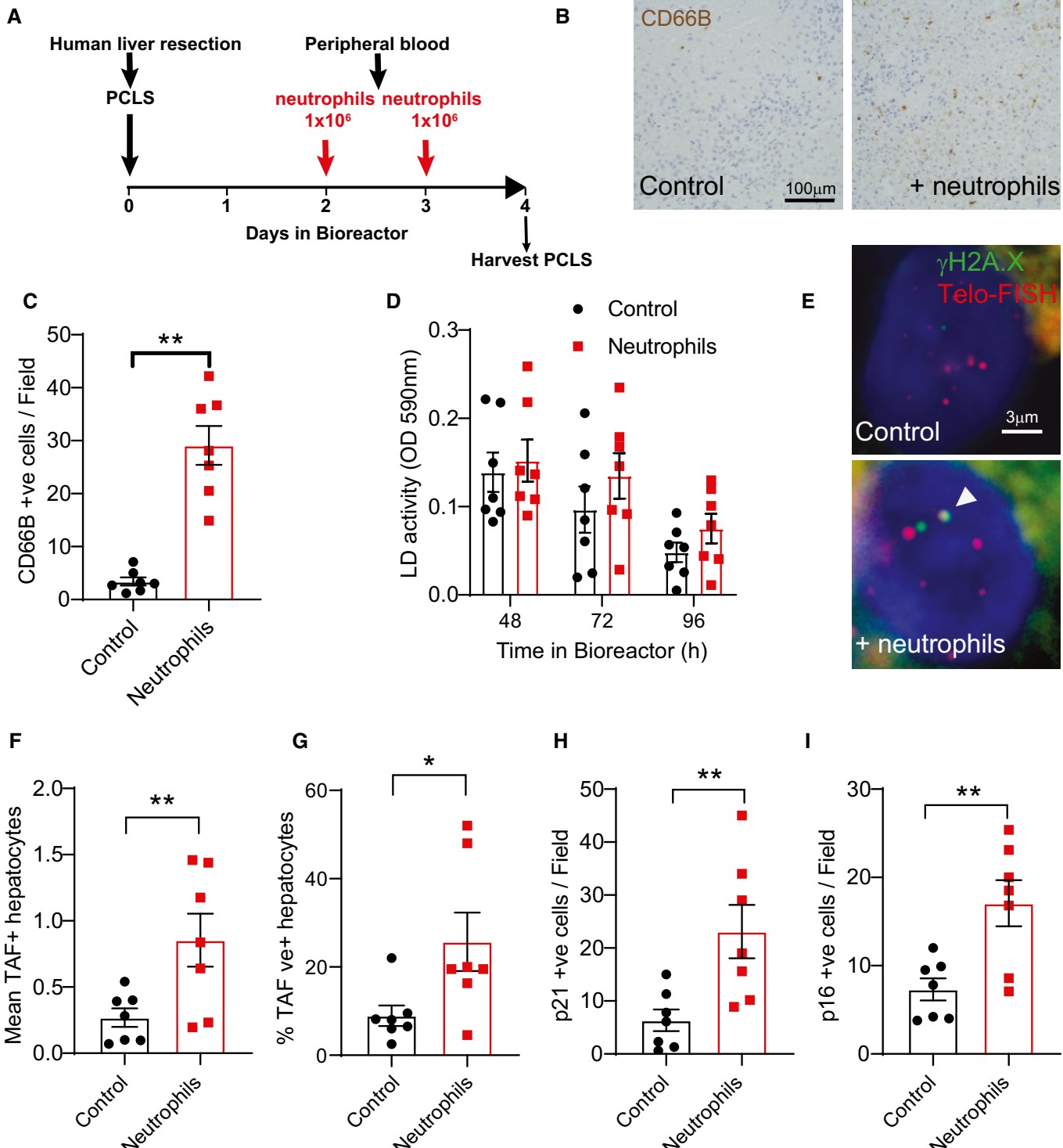

**Figure 3.**

**Figure 3.  Neutrophils induce telomere dysfunction and senescence in human precision-cut liver slices (PCLS).**

A  Schematic depicting experimental design.
B  Representative micrographs of neutrophil marker CD66B.
C  Quantification of neutrophil counts *per* field.
D  Lactate dehydrogenase (LD) activity as a function of time in bioreactor.
E  Representative Immuno-FISH micrographs (using telomere-specific (CCCTAA) peptide nucleic acid probe and anti-γH2A.X antibody). Arrow indicates co-localization between telomeres and γH2A.X.
F  Mean number of TAF in hepatocytes.
G  % of TAF-positive hepatocytes.
H  Number of p21-positive hepatocytes per field.
I  Number of p16$^{Ink4a}$-positive hepatocytes per field.

Data information: Data are mean $\pm$ SEM from liver biopsies obtained from seven male patients aged between 49 and 72 years. Neutrophils were isolated from healthy individuals aged between 21 and 30 years (with purity above 90%). Statistical analysis was performed using one-way ANOVA (Holm–Sidak method) for multiple comparisons and two-tailed *t*-test for single comparisons. **$P < 0.01$; *$P < 0.04$.

We next evaluated whether neutrophil depletion (using neutrophil neutralizing antibody against Ly6G) can reduce telomere dysfunction in the context of chronic liver damage induced by NASH (choline deficient high-fat diet feeding). Notably, we found a significant reduction in TAF-positive hepatocytes in neutrophil neutralizing antibody against Ly6G-treated mice compared to controls (Fig EV4L–N), however no significant changes in expression of p21 (Fig EV4O and P). Our data suggest that neutrophils can be drivers of senescence-associated phenotypes in the liver.

### 5- p16$^{Ink4a}$-positive cells recruit neutrophils during aging

Analysis of RNA-sequencing data from livers collected from 3-, 15-, and 24-month-old mice revealed that several genes associated with neutrophil recruitment were up-regulated with age (Fig 6A). Interestingly, we found that both neutrophil infiltrates and TAF-positive hepatocytes increased with age (Fig 6B and C). We also observed in aged mice that hepatocytes located in close proximity to neutrophils displayed higher TAF numbers and that TAF inversely correlated with distance from neutrophils (Fig 6D and E). A less pronounced decrease in total number of γH2A.X foci as a function of distance from neutrophils was also observed (Fig EV5A).

This observation led us to propose the hypothesis that a vicious cycle occurs between neutrophil infiltrates and hepatic senescence. We hypothesize that senescent cells contribute to some extent to the recruitment of neutrophils during aging and that neutrophil recruitment possibly amplifies senescence in the liver. In order to investigate if senescent cells can recruit neutrophils, we first collected conditioned media from young (early PD) and replicatively senescent fibroblasts and measured the chemotaxis of neutrophils using the sub-agarose method (Fig 6F). Our data indicate that neutrophils migrate preferentially toward senescent cells (Fig 6G). N-Formylmethionyl-leucyl-phenylalanine (fMLP) chemoattractant was used as a positive control.

To further investigate this hypothesis *in vivo*, we used the *INK-ATTAC* mouse model, in which a small molecule, AP20187 (AP), induces apoptosis through dimerization of FKBP-fused Casp8, allowing clearance of p16$^{Ink4a}$ senescent cells (Baker *et al*, 2011). We aged INK-ATTAC mice until they were 27 months old and treated them with AP for 2 months (Fig 6H). As expected, we found that aged INK-ATTAC mice had increased mRNA levels of p16$^{Ink4a}$ compared to young mice and this was significantly reduced by addition of AP (Fig 6I). AP treatment of old mice also led to a significant reduction in the frequency of TAF-positive hepatocytes (Figs 6J and EV5B) but no

changes in p21 mRNA expression (Fig EV5C). Consistent with a role for senescent cells in the recruitment of neutrophils, we found that clearance of p16$^{Ink4a}$ cells reduced significantly the number of neutrophils infiltrating in the liver (Fig 6K). It remains a likelihood that AP is clearing p16$^{Ink4a}$-expressing neutrophils directly and this could explain our observations. In order to investigate this, we isolated intrahepatic leukocytes from young and old mouse livers and analyzed them by cytometry by time-of-flight (CyTOF), which allows for mapping and discriminating between different immune cell types (Figs 6L, and EV5D and E, and Appendix Fig S3). We found that while p16$^{Ink4a}$ was expressed in other immune cells such as monocyte-derived macrophages, macrophages, B cells, and dendritic cells, it was not detectable in liver neutrophils (Fig 6M). In our experiment, we found that the % of p16$^{Ink4a}$ increased in dendritic cells and decreased in B cells with age (Fig 6N). Consistent with our results, analyses of published single-cell RNA-sequencing (scRNA-seq) from aged rats (Ma *et al*, 2020) and mice (Mogilenko *et al*, 2020) revealed that while the frequency of neutrophils increased with age in the liver, these had negligible expression of p16$^{Ink4a}$ (Fig EV5F).

In order to further investigate the role of hepatic neutrophil recruitment in the induction of hepatocyte senescence, we injected 3-month-old mice with recombinant murine Cxcl1, a chemokine involved in neutrophil recruitment (Sawant *et al*, 2016) and the SASP (Coppé *et al*, 2010) (Fig 6O). We found that Cxcl1 injection increased the frequency of neutrophil infiltrates in the liver (Fig 6P) as well as the % of TAF-positive hepatocytes (Fig 6Q). Similar results were obtained following injection with lipopolysaccharide (LPS), a potent activator and chemoattractant of neutrophils. LPS resulted in a significant increase in the mean number of neutrophils as well as TAF 24 h after injection (Fig EV5G–J). Interestingly, we found a positive correlation between TAF-positive hepatocytes and neutrophil infiltrates (Fig 6R).

Altogether, our results support the hypothesis that while neutrophils can trigger telomere dysfunction in hepatocytes, once senescent cells are present in the liver we speculate that these may mediate further neutrophil recruitment and exacerbate telomere dysfunction.

## Discussion

Neutrophils, because of their primary functions in the fight against pathogens, contain pore forming molecules, hydrolytic, and oxidative compounds that can inadvertently cause serious secondary

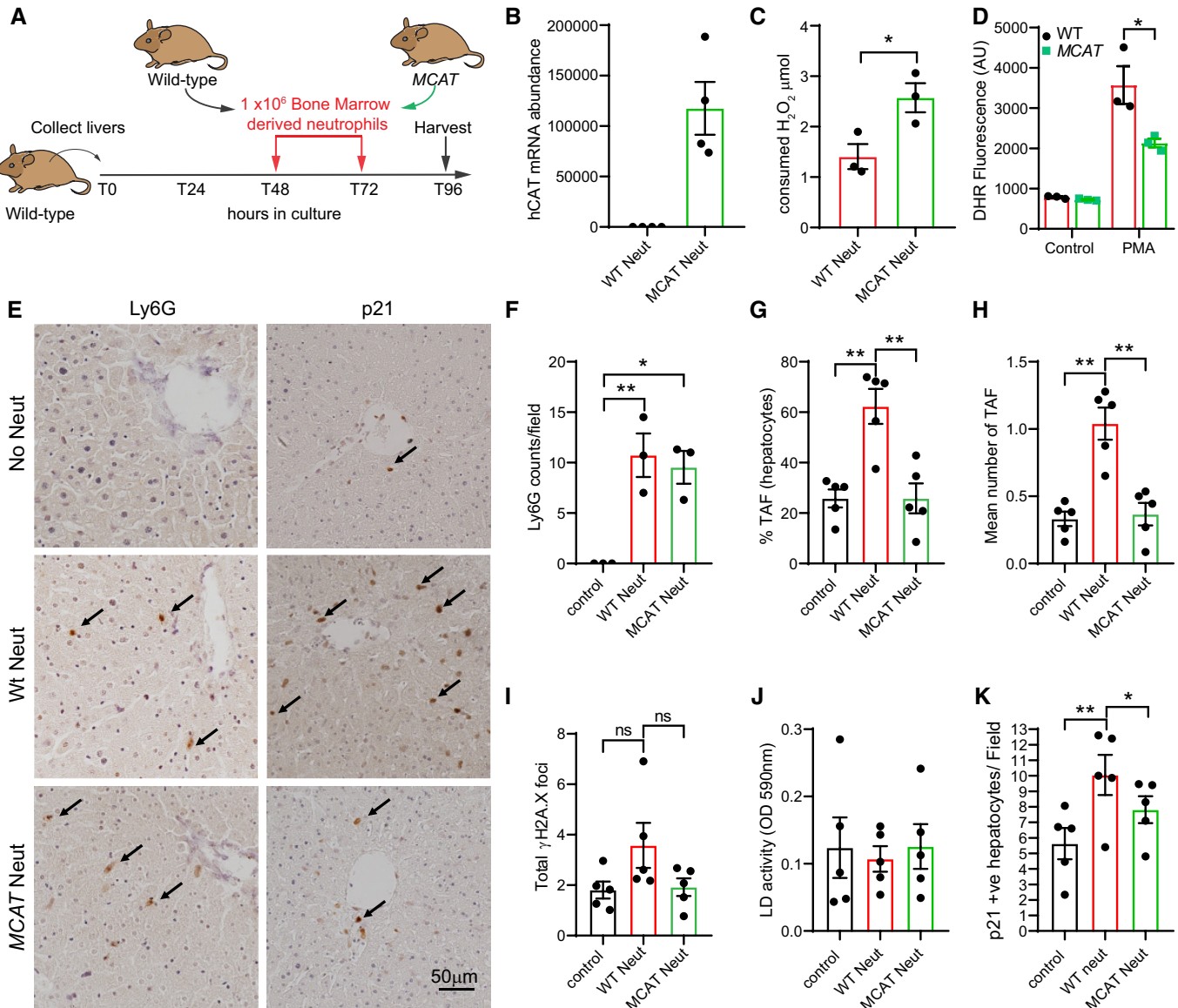

**Figure 4. Neutrophils induce telomere dysfunction in a ROS-dependent manner in precision-cut liver slices (PCLS).**

A Schematic depicting experimental design.
B mRNA expression of hCAT in bone marrow-derived neutrophils from wild-type and *MCAT* mice.
C Catalase activity in bone marrow-derived neutrophils from wild-type and *MCAT* mice.
D Dihydrorhodamine 123 fluorescence was assessed by flow cytometry in neutrophils from wild-type and *MCAT* mice with or without stimulation with PMA.
E Representative micrographs of neutrophil marker Ly6G and senescent-associated marker p21 (96 h).
F Number of Ly6G-positive cells per field (96 h).
G % of TAF-positive hepatocytes (96 h).
H Mean number of TAF in hepatocytes (96 h).
I Total number of γH2A.X foci in hepatocytes (96 h).
J Lactate dehydrogenase (LD) activity (96 h).
K p21-positive cells per field (96 h).

Data information: Data are mean ± SEM of 3–5 independent PCLS. Statistical analysis was performed using one-way ANOVA (Holm–Sidak method) for multiple comparisons and two-tailed *t*-test for single comparisons. **$P < 0.01$; *$P < 0.04$.

damage to the microenvironment (Segal, 2005). Thus, augmented recruitment of neutrophils to sites of sterile inflammation can be detrimental and be a contributor to various diseases (Németh *et al*, 2020). However, the relationship between neutrophils and cellular

senescence, a key driver of aging and disease, remains relatively understudied.

Studies have shown that senescent cells communicate with the immune system and mediate their own clearance. For instance, it

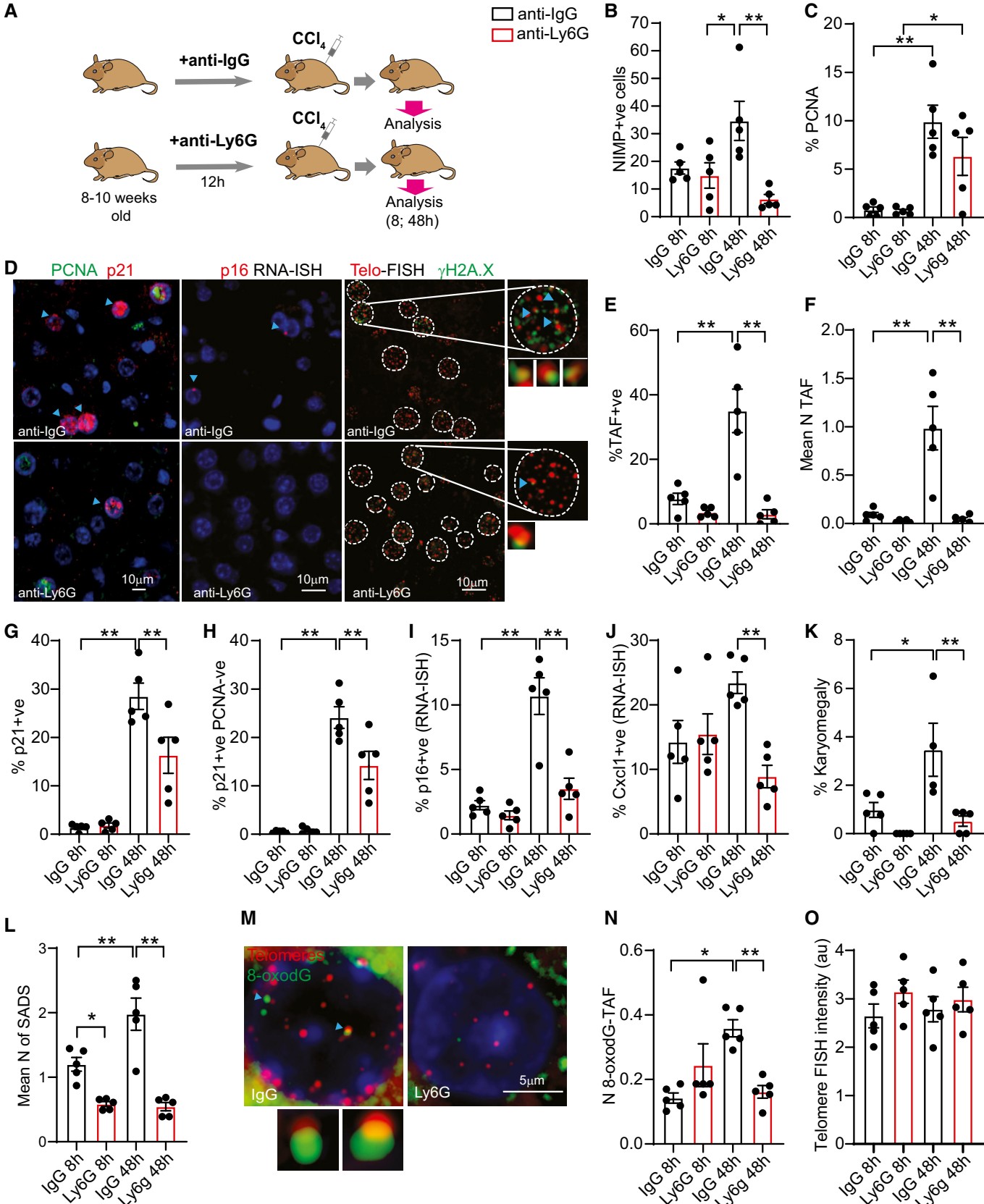

Figure 5.

◀

**Figure 5. Neutrophils induce telomere dysfunction and senescence-associated pathways *in vivo* in a model of acute liver injury.**

A   Schematic depicting experimental design. Mice were pre-treated with neutrophil neutralizing antibody against Ly6G (or with IgG control) for 12 h before CCl$_4$ injection (single intraperitoneal injection of CCl$_4$ at a dose of 2 μl/g body weight) and were sacrificed 8 h and 48 h after injection.
B   Quantification of neutrophils (NIMP+ve) in liver.
C   Quantification of % of PCNA-positive hepatocytes.
D   Representative micrographs of PCNA and p21 immunofluorescence, p16 RNA in situ hybridization (RNA-ISH) and Immuno-FISH (telomere FISH and γH2A.X).
E   % of TAF-positive hepatocytes.
F   Mean number of TAF in hepatocytes.
G   p21-positive hepatocytes.
H   % p21-positive PCNA negative hepatocytes.
I   % p16 mRNA-positive hepatocytes.
J   % Cxcl1 mRNA-positive hepatocytes.
K   % of karyomegalic hepatocytes.
L   Mean number senescence-associated distension of satellites (SADS).
M   Representative Immuno-FISH (telomere FISH and 8-oxodG).
N   Mean number of 8-oxodG co-localizing with telomeres in hepatocytes.
O   Mean telomere FISH intensity in hepatocytes.

Data information: Data are mean ± SEM of five mice *per* group. Statistical analysis was performed using one-way ANOVA (Holm–Sidak method) for multiple comparisons and two-tailed *t*-test for single comparisons. \*\**P* < 0.01; \**P* < 0.04.

was shown that during chronic liver damage, NK cells are involved in the clearance of senescent stellate cells (Krizhanovsky *et al*, 2008). In addition, it has been shown that pre-malignant senescent hepatocytes mediate their own clearance *via* the action of CD4 (+) T cells, monocytes, and macrophages (Kang *et al*, 2011). Interestingly, depletion of neutrophils did not affect significantly senescence surveillance in this model (Kang *et al*, 2011). Thus, while neutrophils may be present in close proximity to senescent hepatocytes, the data available to date suggest that they are not involved directly in their clearance.

In this work, we hypothesized that neutrophil recruitment to sites of sterile inflammation could be a contributor to cellular senescence. Our *in vitro and ex vivo* data indicate that ROS released by neutrophils can accelerate telomere dysfunction and thus induce premature senescence. Furthermore, our data indicate that neutrophil-induced paracrine senescence requires cell-to-cell contact, since human fibroblasts exposed to conditioned media from neutrophils or co-cultured with neutrophils in transwells did not show premature senescence. These findings are reminiscent of a previous report, which suggested that paracrine senescence is ROS-dependent and requires cell-to-cell contact through gap junctions (Nelson *et al*, 2012). Neutrophils are capable of producing and secreting a number of pro-inflammatory factors, some of which have been implicated in the induction of senescence in a non-autonomous fashion (Acosta *et al*, 2013). While our data indicate that it is unlikely that longer-lived soluble factors released by neutrophils are major contributors to paracrine senescence, it remains a possibility that these may be transmitted via gap junctions or tunneling nanotubes (Ariazi *et al*, 2017).

Telomeres serve as protective structures at the ends of chromosomes, and previous work has revealed they are highly susceptible to oxidative stress mostly due to the high content of guanine residues, which can easily undergo oxidative modifications (Oikawa *et al*, 2001). Accordingly, mild oxidative stress has been shown to induce single-stranded breaks preferentially at telomeres and accelerate the rate of telomere shortening (Petersen *et al*, 1998; von Zglinicki *et al*, 2000). Induction of 8-doxoG (a common oxidative DNA lesion) specifically at telomere regions accelerates loss of telomeric repeats and reduces proliferation (Fouquerel *et al*, 2019). Oxidative stress may also disrupt the binding of certain shelterin

proteins that protect telomeres (Opresko *et al*, 2005). Additionally, studies show that telomeres are less efficiently repaired than the bulk of the genome, mostly because shelterin components such as TRF2 inhibit DNA-PL and Ligase-IV-mediated non-homologous end joining (NHEJ) (Bae & Baumann, 2007; Fumagalli *et al*, 2012). Telomeres may also "sense" pro-inflammatory factors. For instance, recent studies have shown that pro-inflammatory factors released by senescent cells such as IP-10 and TGF-β activate signaling pathways that induce telomere dysfunction in neighboring cells (Razdan *et al*, 2018; Victorelli *et al*, 2019). Altogether, these studies support the hypothesis that telomeres are sensors of intrinsic and extrinsic stress and may therefore be particularly susceptible to the close proximity of neutrophils.

Supporting a role for telomeres in neutrophil-induced paracrine senescence, we found that co-culture of neutrophils with fibroblasts ectopically expressing hTERT did not affect telomere shortening, cell proliferation, or expression of senescence markers. We have however also observed that DNA damage foci present outside telomere regions are also significantly reduced, which could be suggestive of hTERT having non-canonical effects uncoupled from its role in telomere maintenance (e.g., reducing DNA damage induction or improving overall DNA repair). Telomerase can shuttle from the nucleus to the mitochondria upon oxidative challenge, where it affects mitochondrial function and reduces ROS generation (Ahmed *et al*, 2008). While we did not observe any relocation of telomerase to the cytosol upon neutrophil co-culture, we cannot discard the possibility that telomerase may be conferring resistance to neutrophil-mediated paracrine damage via its non-canonical roles. We have previously shown that upon telomere damage and activation of the DDR, cells produce high levels of ROS which contribute to secondary DNA damage (predominantly in non-telomeric regions) (Passos *et al*, 2010). Thus, it is possible that hTERT, by preventing telomere shortening and induction of TAF, is also reducing non-telomeric DNA damage induced by secondary ROS.

In the context of the liver, hepatocytes containing dysfunctional telomeres have been shown to increase during aging in mice (Hewitt *et al*, 2012) and also in alcoholic liver disease (ALD) (Wilson *et al*, 2015) and non-alcoholic fatty liver disease (NAFLD) (Ogrodnik *et al*, 2017) patients. Furthermore, TAF in hepatocytes were shown

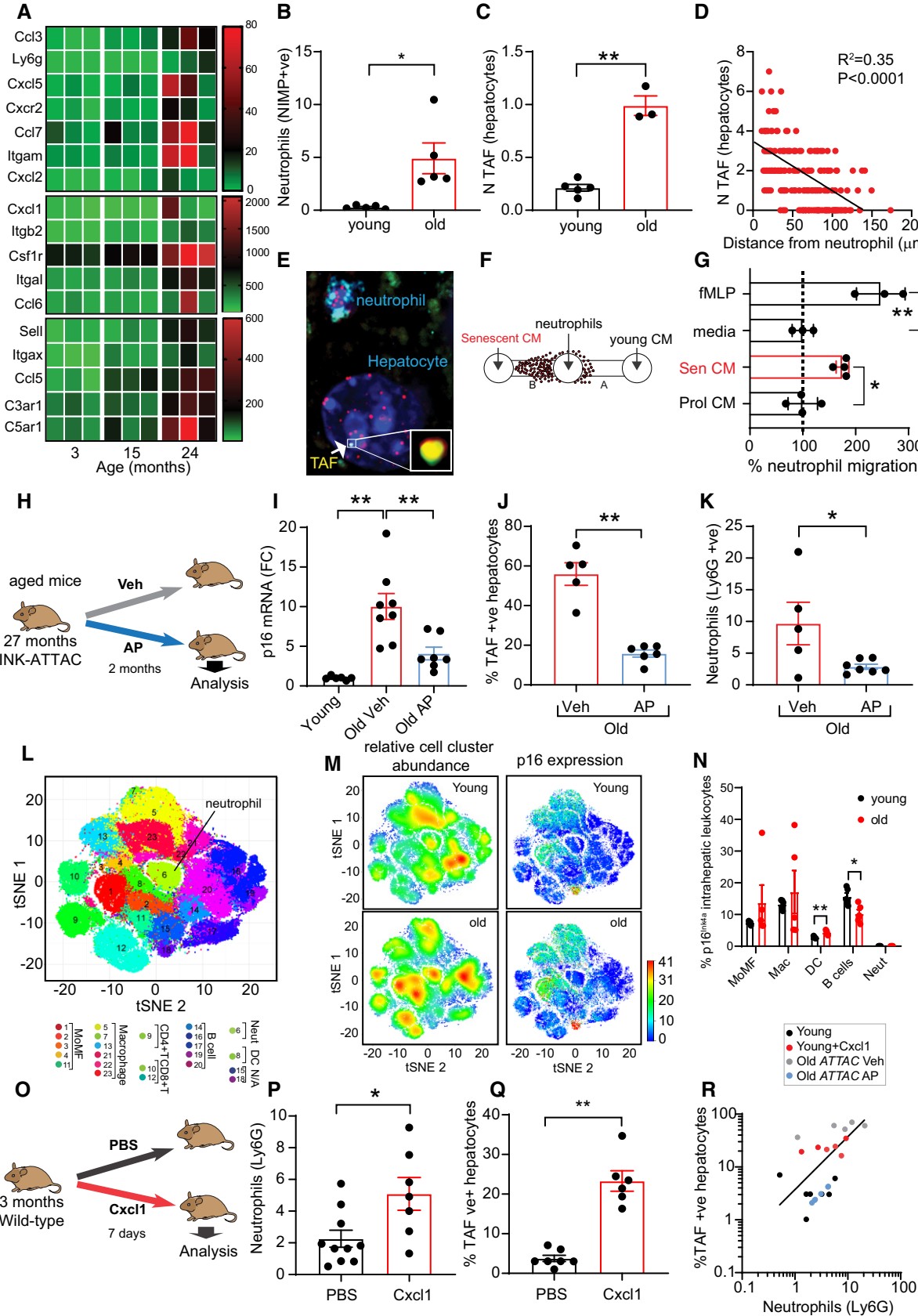

**Figure 6. p16$^{Ink4a}$-expressing cells recruit neutrophils during aging.**

A  RNA-sequencing data showing expression of genes associated with neutrophil recruitment during liver aging.
B, C  Quantification of (B) neutrophil infiltrates (NIMP-positive) and (C) mean number of TAF-positive hepatocytes in liver of young (3 months) and old (24 months) mice. Data are mean $\pm$ SEM of 3–5 mice *per* group.
D  Quantification of mean number of TAF in hepatocytes as a function of the distance from a Ly6G labeled neutrophil. Data are from five mice aged 29 months of age; line indicates linear relationship between mean TAF and distance (linear regression and *F* test: $R^2 = 0.35$; $P < 0.0001$).
E  Representative micrograph showing hepatocyte from aged mouse containing a TAF located in close proximity to a neutrophil (labeled with Ly6G).
F  Schematic depicting experimental design.
G  % of neutrophil migration toward conditioned media (CM) obtained from young (early passage) and replicatively senescent human fibroblasts. Media alone and fMLP were used as negative and positive controls respectively. Data are mean $\pm$ SEM of three independent experiments.
H  Schematic depicting experimental design for panels I–K.
I  Comparison between p16$^{Ink4a}$ mRNA levels of young (3 months old) and old *INK-ATTAC* mice (28–29 months old) treated with vehicle or AP20187. Data are mean $\pm$ SEM of $n = 5$–8 mice *per* age group.
J  % of TAF-positive hepatocytes in old *INK-ATTAC* mice (28–29 months old) treated with vehicle or AP20187. Data are mean $\pm$ SEM of $n = 5$–6 mice *per* group.
K  Liver neutrophils (Ly6G) in old *INK-ATTAC* mice (28–29 months old) treated with vehicle or AP20187. Data are mean $\pm$ SEM of $n = 5$–7 mice *per* group.
L  Intrahepatic leukocyte profiling by mass CyTOF. CyTOF was performed on young (6 months) and old (24 months) *INK-ATTAC* mice ($n = 5$ mice per group); 23 unique clusters of intrahepatic leukocytes were defined by a 24-cell surface marker panel using the Rphenograph clustering algorithm and were visualized on a tSNE plot (t-distributed stochastic neighbor embedding plot).
M  (left) tSNE plots comparing the distribution of young and old intrahepatic leukocytes. Red indicates high frequency categorization of cells to a cluster while blue indicates low frequency; (right) tSNE plots comparing p16$^{Ink4a}$ expression in young and old intrahepatic leukocytes. Red indicates high p16$^{Ink4a}$ expression while blue indicates low expression. Data are from $n = 5$ mice per group.
N  % of p16$^{Ink4a}$-positive cells (detected by CyTOF) in different intrahepatic leukocyte populations isolated from young and old mice. Data are mean $\pm$ SEM of $n = 5$ mice per group.
O  Schematic depicting experimental design for panels P and Q. 3-month-old wild-type mice were injected with murine Cxcl1 for 7 days.
P  Quantification of liver neutrophils (Ly6G) in mice injected with vehicle or Cxcl1. Data are mean $\pm$ SEM of $n = 7$–10 mice *per* group.
Q  Quantification of the % of TAF-positive hepatocytes in mice injected with vehicle or Cxcl1. Data are mean $\pm$ SEM of $n = 6$–7 mice *per* group.
R  Correlation between TAF-positive hepatocytes and neutrophils (linear regression and *F* test: $r^2 = 0.70$; $P = 0.0001$).

Data information: Statistical analysis was performed using one-way ANOVA (Holm–Sidak method) for multiple comparisons and two-tailed *t*-test for single comparisons. **$P < 0.01$; *$P < 0.04$.

to be enhanced by factors such as chronic inflammation (Jurk *et al*, 2014), obesity (Ogrodnik *et al*, 2017), oxidative stress (Jurk *et al*, 2014), and impaired autophagy (Cassidy *et al*, 2020). Importantly, genetic and pharmacological clearance of senescent cells has been shown to reduce TAF in hepatocytes, indicating that it is a robust indicator of cellular senescence. While our data support that neutrophils can induce TAF in hepatocytes, it is difficult to discern what is cause or consequence in the process. Chemotaxis assays show that neutrophils are preferentially recruited by senescent cells and clearance of p16$^{Ink4a}$-positive cells in aged livers from *INK-ATTAC* mice reduces neutrophil infiltrates. However, our data do not allow us to conclusively identify the p16$^{Ink4a}$-positive cell types in the liver that contribute to neutrophil migration. In the case of *INK-ATTAC* mice, AP kills cells based on high expression of p16$^{Ink4a}$, which may include activated immune cells such as macrophages. We have however shown by CyTOF and scRNA-seq analysis that infiltrating neutrophils do not express p16$^{Ink4a}$ in aged liver, suggesting that their reduction following AP treatment is most likely a consequence of elimination of other cell types. Injection of chemoattractant Cxcl1 or LPS led to neutrophil recruitment and induction of TAF; however, we acknowledge that this does not allow us to conclude a causal relationship between both, particularly since these factors may induce senescence directly or stimulate the recruitment of other immune cells apart from neutrophils that can conceivably induce paracrine effects.

Thus, while our results are consistent with a model in which the appearance of a few senescent cells may transmit senescence to otherwise healthy cells *via* the recruitment of neutrophils, this remains to be tested experimentally.

Our study supports the hypothesis of an evolutionary trade-off between efficacy of the immune system and age-related pathology.

While neutrophils have evolved to play critical roles in microbial killing and inflammation, they may contribute to collateral damage and induction of cellular senescence which will be detrimental later in life.

# Materials and Methods

### Animals and procedures

Wild-type mice were inbred C57BL/6 (Harlan, Blackthorn UK). Mice were housed in same-sex cages in groups of 4–6 (56 × 38 × 18 cm, North Kent Plastics, Kent, UK) and individually identified by an ear notch. Mice were housed at 20 $\pm$ 2°C under a 12 h light/12 h dark photoperiod with lights on at 7.00 am. All work complied with the guiding principles for the care and use of laboratory animals and was licensed by the UK Home Office.

### *Acute liver injury using carbon tetrachloride (CCl$_4$)*

Single intraperitoneal injection of CCl$_4$ at a dose of 2 μl (CCl4: olive oil, 1:1 [v:v])/g body weight was administered for 8 and 48 h to 8- to 10-week-old male littermates. Mice were pre-treated with neutralizing antibody against Ly6G or IgG control antibody for 12 h before CCl$_4$ injection. Animals were culled at 48 h post-CCl$_4$ injection.

### *p16$^{Ink4a}$ clearance*

Experimental procedures were approved by the Institutional Animal Care and Use Committee at Mayo Clinic (protocol A26415). *INK-ATTAC* transgenic mice were generated and genotyped as previously described (Baker *et al*, 2011) based on experimental strategies devised by J.L.K., T.T., J. van Deursen and D. Baker at Mayo Clinic.

Briefly, *INK-ATTAC mice* were produced and phenotyped at the Mayo Clinic. Controls for the *INK-ATTAC* experiments were *INK-ATTAC*-null C57BL/6 background mice raised in parallel. Mice were housed 2–5 mice *per* cage, at $22 \pm 0.5C$ on a 12–12 h day–night cycle and provided with food and water *ad libitum*. Cages and bedding (autoclaved Enrich-o'Cobs (The Andersons Incorporated)) were changed once *per* week. In all the experiments, littermates of the same sex were randomly assigned to experimental groups. *INK-ATTAC* mice were injected intraperitoneally (i.p.) with AP20187 (10 mg/kg) (MCE MedchemExpress; Cat: # HY-13992/CS1953) or vehicle (4% ethanol, 10% PEG-400, and 2% Tween-20 in distilled water) for 3 days every 2 weeks for a total of 8–10 weeks.

### I.P. Injection of recombinant Cxcl1

Recombinant CXCL1 (Peprotech, #250-11) or vehicle (PBS) was administered to lean C57BL/6 *via* i.p. injection (5 mg/kg in PBS) daily for 7 days.

### Mouse tissue collection and preparation

Livers were collected during necropsy and fixed with 4% formaldehyde aqueous solution (VWR; Cat. Number 9713.9010) and paraffin-embedded for histochemical analysis, sectioned at 3 μm and mounted on Superfrost Plus glass slides. A fraction of the liver was snap-frozen in liquid nitrogen and stored at $-80°C$ for biochemical analyses.

### Cell culture and treatments

Human embryonic lung MRC5 fibroblasts were obtained from ECACC (Europe Collection of Cell Culture) (Salisbury, UK). MRC5 fibroblasts were used for molecular and cellular biology analysis at a population doubling (PD) range of 16–25 for stress-induced senescence or as a starting point for reaching replicative exhaustion (replicative senescence). MRC5 fibroblasts were transfected retrovirally with the human catalytic subunit (TERT) of the enzyme telomerase. Cells were cultured in Dulbecco's modified Eagle's medium (DMEM), supplemented with 10% heat inactivated fetal bovine serum (FBS) (Biosera, Ringmer, UK), 100 μg/ml streptomycin, 100 units/ml penicillin and 2 mM L-glutamine, incubated in a humidified atmosphere at 37°C with 95% air and 5% $CO_2$ (complete medium, classic conditions).

### Neutrophil isolation

Polymorphonuclear leukocytes (PMN) were isolated from peripheral blood of healthy donors. Briefly, after centrifugation of citrated whole blood at 300 *g* for 20 min and removal of platelet-rich plasma, leukocytes were separated from erythrocytes by dextran sedimentation using 0.6% dextran T500. PMN were then separated from mononuclear leukocytes using discontinuous isotonic Percoll gradients. PMN leukocytes were 95–98% neutrophils using morphological criteria, and viability was assessed by trypan blue exclusion.

### Murine neutrophil isolation

Polymorphonuclear leukocytes (PMN) were isolated from the bone marrow of 8 to 12-week-old male C57Bl6/J mice. Briefly, the bone marrow was flushed from the femur and tibia of both hind limbs using 5% FCS HBSS. The single-cell suspension was then washed and placed onto a 62% Percoll Gradient and centrifuged at 1,000 *g*

for 30 min. The pellet containing polymorphonuclear cells were collected, washed, and then counted. Neutrophil purity was assessed by Ly6G and CD11b (BioLegend) flow cytometry (BD FACS Canto II).

### Human neutrophil and fibroblast direct co-culture

MRC5 fibroblasts at an early PD (16–26) were used. Freshly isolated neutrophils were primed or not with 100 ng/ml LPS for 1 h at 37°C, 3% $O_2$ in the dark. After 1 h incubation, neutrophils were centrifuged at 1,000 RPM to wash remaining LPS and $1 \times 10^6$ neutrophils were added to the MRC5 fibroblast culture at a ratio of 1 fibroblast for five neutrophils. Recombinant catalase (100 UI/ml) was added to the suspension for the duration of the co-culture. Every 24 h, neutrophils were removed by aspiration followed by a wash with pre-warmed media and then fresh neutrophils from a different healthy donor were added for a further 24-h co-incubation. This procedure was repeated for up to 3 days; then on last day, neutrophils were removed and the cells were transferred into new T150 flasks and cultured until replicative senescence. During the co-culture, cells were cultivated at 3% $O_2$ in order to better maintain neutrophil viability.

### Indirect co-culture

Neutrophils were first incubated for 1 h with 100 ng/ml LPS at 37°C, 3% $O_2$ in the dark, then centrifuged at 1,000 RPM to wash remaining LPS and resuspended in media and co-cultured with human MRC5 fibroblasts using a 3-μm transwell insert (VWR® Tissue Culture Plate Inserts, Polyester (PET) Membrane, Sterilized, Standard Line) at a 1:5 ratio for 24 h. Every 24 h, fresh neutrophils from a different healthy donor were added to the transwell for a further 24 h (procedure was repeated three times—total 36 h). Separately, primed neutrophils were cultured in media, 3% $O_2$ for 24 h; then, centrifuged and conditioned media (CM) was added to MRC5 fibroblasts. Every 24 h, CM from a different neutrophil healthy donor was added to MRC5 fibroblasts (procedure was repeated three times—total 36 h). Following 3 days of indirect co-culture, cells were transferred into new T150 flasks and cultured until replicative senescence.

### Neutrophil chemotaxis

Neutrophil chemotaxis was evaluated using the sub-agarose method. As a positive control, formyl-methionine-leucine-phenylalanine was used.

### Metaphase spread generation

Metaphase spreads from MRC5 fibroblasts were prepared at different time points by treatment of subconfluent cells with 10 μg/ml colcemid for 24 h at 37°C, followed by 60 mM KCl for 15 min at RT and fixation in ethanol:acetic acid (3:1).

### Quantitative RT–PCR

RNA was extracted using the RNeasy Mini Kit (Qiagen, 74106). Complementary DNAs were synthesized using the High Capacity cDNA Reverse Transcription Kit (Thermo Fisher, 4368814) following the manufacturer's instructions. Quantitative real-time PCR was carried out using PerfeCTa qPCR ToughMix (Quantabio) in a C100TM Thermal Cycler, CFX96TM Real-Time PCR System (Bio-Rad), and Bio-Rad CFX Manager software. Predesigned primers and probes

from IDT PrimeTime. Mouse: p16: Mm.PT.58.42804808; p21: Mm.PT.58.5884610; TBP: Mm.PT.39a.22214839; Human: IL-6: Hs.PT.58.40226675; IL-8: Hs.PT.58.39926886.g; p21: Hs.PT.58.38492863.g; p16: Hs00923894_m1; TBP : Hs.PT.39a.22214825.

### Immuno-cyto and Immunohistochemistry

Cells grown on coverslips were fixed in 2% paraformaldehyde in PBS for 5 min. Cells were then permeabilized in PBG-Triton (PBS, 0.4% Fish-skin gelatin, 0.5% BSA, 0.5% Triton X-100) for 45 min and incubated with primary antibody overnight at 4°C. Following PBS washes, cells were incubated with secondary antibody for 45 min and mounted onto glass microscope slides with ProLong Gold Antifade Mountant with DAPI (Invitrogen).

For FFPE tissues, sections were deparaffinized in 100% Histoclear, hydrated in 100, 90, and 70% ethanol and incubated twice for 5 min in distilled water. Antigen retrieval was performed by incubating sections in 0.01 M citrate buffer (pH 6.0) and heated until boiling for 10 min. Sections were allowed to cool down to room temperature followed by two washes in distilled water for 5 min. Next, sections were blocked in normal goat serum (1:60) in BSA/PBS for 30 min and incubated with primary antibody overnight at 4°C. Following three PBS washes, sections were then incubated with secondary antibody for 1 h, followed by PBS washes and mounted using ProLong Gold Antifade Mountant with DAPI (Invitrogen).

### Antibodies

Primary antibodies used were as follows: mouse monoclonal anti-p16 (dilution as provided by manufacturer, 9511; CINTec Histology, Roche) and rabbit polyclonal anti-Ki67 antibody (1:1,000;ab15580; 4 μg/ml Abcam), Anti-phospho-Histone H2A.X (Ser139) Antibody, clone JBW301 (1:1,000), Anti-PCNA antibody [PC10] (ab29) mouse (1:1,000, Abcam), NIMPR-14 rat anti-mouse monoclonal (1:200 Abcam); Anti-p21 rat antibody [HUGO291] (ab107099); Anti-p21 rabbit antibody (1:100 Abcam); Ly6g (BP0075-1) rat monoclonal (1:100, BioXcell); 8oxoDG Anti-8-Oxoguanine Antibody, clone 483.15, Merck Millipore, MAB3560 (mouse 1:200); CD66B antibody (1:200, 305102—BioLegend); rabbit anti-telomerase catalytic subunit (hTERT) (1:500, Rockland, 600-401-252S).

### For Western blot

Rabbit Monoclonal (12D1) anti-p21 Waf1 Cip1 (1:1,000, 2947S, Cell Signalling), Rabbit Monoclonal (D3W8G) anti-p16 INK4A (1:1,000, 92803—Cell Signalling), Rabbit Polyclonal anti-p15 CDKN2B (1:500, SAB4500078—Sigma-Aldrich), Rabbit Polyclonal anti-β-Tubulin (1:1,000, 2146S—Cell Signalling ), Rabbit Polyclonal anti-Lamin B1 (1:10,000, Abcam—ab16048).

### TUNEL staining

TUNEL-HRP-DAB staining (ab206386) was performed as per manufacturer's instructions.

### Immuno-FISH and Q-FISH

For cells grown on coverslips, immunocytochemistry was performed as described above using mouse monoclonal anti-γH2AX (1:200, 05-636; Millipore) or rabbit anti-53bp1 (1:500, # 4937S, Cell Signaling). Following secondary antibody incubation, cells were fixed in methanol: acetic acid (3:1 ratio) for 30 min followed by dehydration in graded cold ethanol solutions (70, 90, 100%) for 2 min each.

Cells were incubated in PBS at 37°C for 5 min and fixed in 4% paraformaldehyde at 37°C for 2 min. Following a PBS wash, cells were dehydrated again with cold ethanol solutions (70, 90, 100%) for 2 min each and left to air-dry. Coverslips were then placed onto glass slides containing 10 μl of PNA hybridization mix (70% deionized formamide (Sigma), 25 mM MgCl₂, 1 M Tris pH 7.2, 5% blocking reagent (Roche) containing 2.5 μg/ml Cy-3-labeled telomere-specific (CCCTAA) peptide nucleic acid probe or FAM-labeled, CENPB-specific (centromere) (ATTCGTTGGAAACGGGA) peptide nucleic acid probe (Panagene), and samples were denatured for 10 min at 80°C. Samples were incubated in a humidified chamber protected from light for 2 h at room temperature to allow hybridization to occur, followed by washes with 70% formamide in 2 × SSC (3 × 10 min each) and three washes in 0.05% TBS-Tween-20 for 5 min. Cells were then dehydrated in graded cold ethanol solutions (70, 90, 100%) and left to air-dry before they were mounted onto glass microscope slides with ProLong Gold Antifade Mountant with DAPI (Invitrogen). Cells were imaged using in-depth Z stacking (a minimum of 35 optical slices with 63× objective).

For FFPE tissues, immunohistochemistry was carried out as described above. Following an overnight incubation with rabbit monoclonal anti-γH2AX (1:200, 9718; Cell Signalling) or 8oxoDG anti-8-oxoguanine antibody, clone 483.15, Merck Millipore, MAB3560 (mouse 1:200), sections were incubated with a goat anti-rabbit biotinylated secondary antibody (1:200, PK-6101; Vector Labs) for 30 min at room temperature. Following three PBS washes, tissues were incubated with fluorescein avidin DCS (1:500, A-2011; Vector Labs) for another 30 min at room temperature. Sections were then washed three times in PBS and cross-linked by incubation in 4% paraformaldehyde in PBS for 20 min. Sections were washed in PBS three times and then dehydrated in graded cold ethanol solutions (70, 90, 100%) for 3 min each. Tissues were then allowed to air-dry prior to being denatured in 10 μl of hybridization mix (as described above) for 10 min at 80°C and then incubated for 2 h at room temperature in a dark humidified chamber to allow hybridization to occur. Sections were washed in 70% formamide in 2 × SCC for 10 min, followed by a wash in 2 × SSC for 10 min, and a PBS wash for 10 min. Tissues were then mounted using ProLong Gold Antifade Mountant with DAPI (Invitrogen). Sections were imaged using in-depth Z stacking (a minimum of 40 optical slices with 63× objective) followed by Huygens (SVI) deconvolution.

Quantitative FISH (Q-FISH) analysis of telomere FISH intensity was performed on fixed cells and FFPE tissue using the same protocol.

### RNA in situ hybridization

RNA-ISH was performed after RNAscope protocol from Advanced Cell Diagnostics Inc. (ACD). Paraffin sections were deparaffinized with Histoclear, rehydrated in graded ethanol (EtOH), and H₂O₂ was applied for 10 min at RT followed by two washes with water. Sections were placed in hot retrieval reagent and heated for 30 min. After washes in water and 100% EtOH, sections were air-dried. Sections were treated with protease plus for 30 min at 40°C, washed with water and incubated with target probe (p16, Cxcl1, Il1-α) for 2 h at 40°C. Afterward, slides were washed with water followed by incubation with AMP1 (30 min at 40°C) and then washed with wash buffer (WB) and AMP2 (15 min at 40°C), WB and AMP3 (30 min at 40°C), WB and AMP4 (15 min at 40°C), WB and AMP5 (30 min at

RT) and WB and, finally, AMP6 (15 min at RT). Finally, RNAscope 2.5 HD Reagent kit-RED was used for chromogenic labeling. Tissues were then mounted using ProLong Gold Antifade Mountant with DAPI (Invitrogen).

### Human precision-cut liver slices

Human liver tissue from the normal margins of colorectal metastasis resections were obtained with full ethical approval (H10/H0906/41) and CEPA biobank (17/NE/0070) and used subject to patients' written consent. All experiments conformed to the principles set out in the WMA Declaration of Helsinki and the Department of Health and Human Services Belmont Report. Precision-cut liver slices were cut using a Leica VT1200S microtome (Leica Biosystems). Slices were then cultured for 4 days at 37°C supplemented with 5% $CO_2$ in BioR plates in our patented bioreactor platform as previously described (Paish et al, 2019). Culture media were collected for analysis and replaced daily with fresh culture media. Human neutrophils were isolated from peripheral blood of healthy donors as described previously, and $1 \times 10^6$ neutrophils were added per tissue slices on days 2 and 3 post-tissue slicing. Tissue slices were harvested at day 4, formalin-fixed, and then paraffin-embedded for future analysis.

### Murine liver slice experiment details

Murine precision-cut liver slices were generated and cultured as previously described (Paish et al, 2019). Briefly, livers of 8- to 12-week-old male C57Bl6/J mice were cored using an 8-mm biopsy punch and embedded in agarose. Precision-cut liver slices were cut using a Leica VT1200S microtome (Leica Biosystems) and cultured for 4 days at 37°C supplemented with 5% $CO_2$ in BioR plates in our patented bioreactor platform. Cell culture medium was collected and replaced daily. Murine neutrophils were isolated as previously described, and $1 \times 10^6$ neutrophils were added per tissue slice on days 2 and 3 post-tissue slicing. Tissue slices were harvested at day 4, formalin-fixed, and then paraffin-embedded for future analysis.

### hCAT expression

RNA was extracted using the QIAGEN RNeasy Mini Kit according to manufacturer's instructions from bone marrow-derived neutrophils isolated from C57BL/6.mCAT mice (MCAT transgenic mice) and WT mice. cDNA was synthesized using the GoScript Reverse Transcription System (Promega). RT–PCR was performed using SYBR Green Jumpstart ready mix and the following primers; Forward—GAGCCTACGTCCTGAGTCTC, Reverse—ATCCCGGATGC CATAGTCAG.

### Catalase activity assay

Catalase Activity Assay Kit (ab83464) was performed as per manufacturer's instructions on bone marrow-derived neutrophils isolated from WT and MCAT mice.

### Lactate dehydrogenase assay

Lactate dehydrogenase (Thermo Fisher) assay kit was performed as per the manufacturer's instructions.

### RNA-sequencing

RNA-seq was carried out as previously described in (Ogrodnik et al, 2017). Briefly, strand-specific paired-end libraries for RNA-seq were generated from DNAse-treated total RNA using Ribozero and ScriptSeq systems (Epicentre/Illumina) and run on an Illumina 2500 sequencer to obtain 100 base pair paired-end reads with four libraries multiplexed per flow cell lane using 100 bp paired-end reads. This resulted in an average of 250 million reads per lane, with an average of 40 million reads per sample. Each individual library received a unique Illumina barcode. Low-quality reads were filtered out using Kraken (Davis et al, 2013). The resulting filtered reads were mapped to the mouse genome version mm10 using STAR aligner (Dobin et al, 2013) followed by estimates of raw gene counts using HTSeq (Anders et al, 2015). Differential gene expression was analyzed using DESeq2 (Love et al, 2014). Statistical significance was expressed as a P value adjusted for a false discovery rate of 0.01 using the Benjamini–Hochberg correction for multiple testing.

### Alkaline comet assay

Cells were trypsinized and frozen in 10% DMSO in fetal bovine serum and stored at −80°C prior to downstream processing. Cells were washed in cold PBS and resuspended in 0.7% Low Melting Point agarose (Sigma, A9414) at 37°C to a concentration of $2 \times 10^5$ cells/ml. Cell/agarose mix (70 μl) was placed on slides previously coated in 1% agarose. Slides were incubated in lysis buffer (2.5 M NaCl, 100 nM EDTA, 10 nM Tris, pH 10, 250 nM NaOH, 10% DMSO, 1% Triton X-100) for 1 h at 4°C. Slides were then washed twice in cold PBS. Samples were subjected to electrophoresis for 40 min at 25 V at 4°C in alkaline buffer (300 mM NaOH, 1 mM EDTA). Slides were then washed twice in cold PBS and stained with Sybr Gold (Life Technologies, S11494) in Tris-borate EDTA buffer for 40 min. Slides were washed twice in MilliQ water and allowed to dry. Samples were imaged using an Olympus BX51 widefield microscope with Olympus UPlanFL 20×/0.50 air objective. Comets were scored using Comet assay IV (Perceptive Instruments). For each sample, 100 randomly captured comets (50 cells on each of 2 comet slides) were quantified.

### Superoxide anion release by neutrophils

The release of superoxide anions by freshly isolated neutrophils in response to fMLP was measured using cytochrome c reduction assay. Cells were pre-incubated with LPS, PMA, or HBSS containing $Ca^{2+}$ and $Mg^{+2}$, or platelet-activating factor (positive control) for 60 min, in the presence of 1% autologous serum, before stimulation with fMLP (100 nM). For each sample, a control with superoxide dismutase (200 U) was included to verify the specificity of cytochrome c reduction by superoxide anions. After 15 min incubation at 37°C, the reaction was stopped immediately by placing the cells on ice then centrifuged at 10,000 g for 3 min. The superoxide dismutase reduction of cytochrome c was then determined for each supernatant by measuring the absorbance at 550 nm using a BMG Fluostar 67. Optima plate reader (BMG Labtech Ltd, Aylesbury, UK). Results are expressed as nmole of $O_2.^-$ per million neutrophils, using the extinction coefficient of $21 \times 10^3 \ M^{-1} \ cm^{-1}$.

### Dihydrorhodamine 123 (DHR)

The neutrophils were incubated in DHR final concentration 1 μM at 37°C for 15 min and then subsequently stimulated with platelet-activating factor (PAF) at 100 nM for 10 min at 37°C. Median fluorescence intensity for DHR was assessed by flow cytometry (BD FACS Canto II).

### Mass cytometry by time-of-flight (CyTOF)

For mass cytometry experiment, we used 6-month-old ("young") and 24-month-old ("old") *INK-ATTAC* mice. Liver tissue (1.6 g) was dissociated using mouse liver dissociation kit (Miltenyi, #130-105-807) according to the manufacturer's instruction, including cell debris removal step (Miltenyi, #130-109-398). Intrahepatic leukocytes were purified by Percoll (Sigma-Aldrich, #P1644) gradient centrifugation and stained as previously described in detail (Guo *et al*, 2019). CyTOF (Fluidigm) mass cytometry and data analysis were performed by the Immune Monitoring Core at Mayo Clinic as previously described (Guo *et al*, 2019). Visualization of different immune cell clusters was mapped by using a tSNE map. Relative marker intensity and cluster abundance for each sample were plotted using a heatmap. The CyTOF antibody panel used is listed in Appendix Table S1.

### Statistical analysis

All statistical analyses were carried out using GraphPad Prism 8.01. For comparisons between two groups, a two-tailed unpaired *t*-test was used, and where the data were not normally distributed, a Mann–Whitney *U*-test was performed. For multi-group comparison, a one-way ANOVA with Tukey's or Holm–Sidak's *post hoc* test was used.

## Data availability

The RNA-seq data from this publication have been submitted to the European Nucleotide Archive https://www.ebi.ac.uk/ena/browser/home, accession numbers: PRJEB42908 and ERP126835.

**Expanded View** for this article is available online.

## Acknowledgements

The authors acknowledge support from NIH grants: R01 AG068048 (JFP), R01 AG068182-01 (DJ), R37 AG013925 (JLK, TT), R01 AG050582 (NN) and P01 AG062413 (SK, JLK, DJ, TT, JFP), the Connor Fund (JLK, TT), Robert J. and Theresa W. Ryan (JLK, TT), and the Noaber Foundation (JLK, TT). F.O and D.A.M are supported by MRC program Grants MR/K0019494/1 and MR/R023026/1. D.A.M is supported by a CRUK program grant, reference C18342/A23390. JFP, DJ, AL, and SV would like to thank the Ted Nash Long Life Foundation, UL1 TR0002377 from the National Centre for Advancing Translational Sciences (NCATS) and BBSRC [grants BB/L502066/1; BB/K017314/1]. DG is supported by the Newcastle CRUK Clinical Academic Training Programme. AC is funded by the W.E. Harker Foundation. PH is supported by the AASLD Foundation. DJ and PH are supported by P30DK084567. CAM was supported by a fellowship from Coordenação de Aperfeiçoamento de Pessoal de Nível Superior—Brasil (Capes)—Finance Code 001. MCAT mice were a kind gift from Prof. Owen Samson at the Cancer Research UK Beatson Institute. We also thank the technical support provided by Dr. Glyn Nelson and the Newcastle University Bioimaging unit.

## Author contributions

AL performed the majority of the experiments; JL, MHRS, SV, PH, MO, AC, LH, SAV, HS, AH, MGV, SE, DG, KP, DD, MD, SG, SG, CW, SP, JB, BC, JC, NN, and CAM performed and evaluated individual experiments; TVZ, GS, HR, FO, SK, TT, JLK, MH, and DJ designed and supervised individual experiments; DAM and JFP designed and supervised the study; JFP and DAM wrote the manuscript with the contributions from all the authors.

## Conflict of interest

FO and DAM are directors and shareholders in Fibrofind limited. JL is a shareholder in Fibrofind limited. Patents on INK-ATTAC mice are held by Mayo Clinic and licensed to Unity Biotechnology. JLK and TT may gain financially from these patents and licenses. This research has been reviewed by the Mayo Clinic Conflict of Interest Review Board and was conducted in compliance with Mayo Clinic Conflict of Interest policies. The remaining authors declare no competing financial interests.

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
