## [Review Process File · The EMBO Journal]

Neutrophils induce paracrine telomere dysfunction and senescence in ROS-dependent manner.

Anthony Lagnado, Jack Leslie, Marie-Hélène Ruchaud-Sparagano, Stella Victorelli, Petra Hirsova, Mikolaj Ogrodnik, Amy Collins, Maria Vizioli, Leena Habiballa, Gabriele Saretzki, Shane Evans, Hanna Salmonowicz, Adam Hruby, Daniel Geh, Kevin Pavelko, David Dolan, Helen Reeves, Sushma Grellscheid, Colin Wilson, Sanjay Pandanaboyana, Madison Doolittle, Thomas von Zglinicki, Fiona Oakley, Suchira Gallage, Caroline Wilson, Jodie Birch, Bernadette Carroll, James Chapman, Mathias Heikenwälder, Nicola Neretti, Sundeep Khosla, Claudio Masuda, Tamara Tchkonja, Jams L. Kirkland, Diana Jurk, Derek Mann, and Joao Passos

DOI: 10.15252/embj.2020106048

Corresponding authors: Joao Passos (Passos.Joao@mayo.edu) , Derek Mann (derek.mann@newcastle.ac.uk)

Review Timeline:

Submission Date:	29th Jun 20
Editorial Decision:	7th Aug 20
Revision Received:	16th Dec 20
Editorial Decision:	26th Jan 21
Revision Received:	9th Feb 21
Accepted:	15th Feb 21

Editor: Daniel Klimmeck

Transaction Report:

Dear Joao,

Thank you again for the submission of your manuscript (EMBOJ-2020-106048) to The EMBO Journal, and in addition providing us with further input in your preliminary revision plan. As mentioned earlier, your study has been sent to three referees, and we have received reports from all of them, which I enclose below.

As you will see, the referees acknowledge the potential interest and novelty of your findings, although they also express major concerns on the analysis. In particular, referee #1 states that the claims made on causalities between neutrophils and hepatocyte senescence (ref#1, pt.5) as well as senescent hepatocytes and neutrophil recruitment (ref#1, pt. 6) are not sufficiently supported by the current data. Further, this referee finds that the depth of senescence characterisation and mechanistic insights into the bystander induction are too preliminary (ref#1, pts. 2,5). Referee #2 is concerned that the claims on telomere involvement in the paracrine effect and cytokines contributing to this induction are not well supported by the data (ref#2, pts.1, 4-6, see also ref#3, A)). In addition, the referees point to a number of issues related to missing controls on potential confounding factors, use of appropriate markers and data inconsistencies.

I judge the comments of the referees to be generally reasonable and given their overall interest, and considering the experimental work outlined in your preliminary response, we are in principle happy to invite you to revise your manuscript to address the referees' comments. I need to stress though that we do need strong support from the referees on a revised version of the study in order to move on to publication of the work and as to the open outcome of the revisional work suggest to keep EMBO Reports in mind for this work as an alternative venue.

Please let me know any time if you have additional questions or need further input on the referee comments.

As discussed, we generally allow three months as standard revision time. As a matter of policy, competing manuscripts published during this period will not negatively impact on our assessment of the conceptual advance presented by your study. However, we request that you contact the editor as soon as possible upon publication of any related work, to discuss how to proceed. Should you foresee a problem in meeting this three-month deadline, please let us know in advance and we may be able to grant an extension.

In this context I also want to point to our adjusted GTA. We are aware that many laboratories cannot function at full efficiency during the current COVID-19/SARS-CoV-2 pandemic and have therefore extended our 'scooping protection policy' to cover the period required for a full revision to address the experimental issues highlighted in the editorial decision letter. Please contact us at any time to discuss an adapted revision plan for your manuscript should you need additional time, and also if you see a paper with related content published elsewhere.

Thank you for the opportunity to consider your work for publication. I look forward to your revision.

Kind regards,

Daniel

Daniel Klimmeck, PhD
Editor
The EMBO Journal

Before submitting your revision, primary datasets (and computer code, where appropriate) produced in this study need to be deposited in an appropriate public database (see <https://www.embopress.org/page/journal/14602075/authorguide#datadeposition>).

The accession numbers and database should be listed in a formal "Data Availability" section (placed after Materials & Method) that follows the model below (see also <https://www.embopress.org/page/journal/14602075/authorguide#availabilityofpublishedmaterial>). Please note that the Data Availability Section is restricted to new primary data that are part of this study.

Data availability

Our journal also encourages inclusion of *data citations in the reference list* to directly cite datasets that were re-used and obtained from public databases. Data citations in the article text are distinct from normal bibliographical citations and should directly link to the database records from which the data can be accessed. In the main text, data citations are formatted as follows: "Data ref: Smith et al, 2001" or "Data ref: NCBI Sequence Read Archive PRJNA342805, 2017". In the Reference list, data citations must be labeled with "[DATASET]". A data reference must provide the database name, accession number/identifiers and a resolvable link to the landing page from which the data can be accessed at the end of the reference. Further instructions are available at <https://www.embopress.org/page/journal/14602075/authorguide#referencesformat>

- a point-by-point response to the referees' comments, with a detailed description of the changes made (as a word file).
- a word file of the manuscript text.
- individual production quality figure files (one file per figure)
- a complete author checklist, which you can download from our author guidelines (<http://emboj.embopress.org/authorguide>).
- Expanded View files (replacing Supplementary Information)

Further information is available in our Guide For Authors:

The revision must be submitted online within 90 days; please click on the link below to submit the revision online before 5th Nov 2020.

Referee #1:

This study aims to understand the relationships between neutrophils and senescent cells. The authors suggest that neutrophils accelerate senescence via ROS-mediated telomere dysfunction. They show correlation between neutrophils and telomere dysfunction in acute liver damage. They also suggest that during ageing senescent cells recruit neutrophils to the liver and spread

senescence to other cells via the recruited neutrophils.

This is a novel study that shows an interesting angle of the interaction of senescent cells with the immune system: recruitment of the neutrophils and subsequent possible senescence spread. The concepts described are very interesting. However, multiple aspects of the study, especially ones related to in vivo experiments are quite preliminary and does not allow reaching the conclusions suggested by the authors. It is necessary to provide more data in regard to presented in vivo experiments in order to get some minimal support to the interesting suggested conclusions.

Specific comments:

1. The authors suggest that the effect of neutrophils on proliferative lifespan is mediated by ROS. However in the data presented the effect of catalase, and in fact neutrophils, was only tested in the presence of LPS, which leads to overstimulation of the neutrophils and might affect MRC5 cells as well. Therefore, in order to support the conclusion suggested by the authors, it is necessary to test the effects of neutrophils without LPS in ALL the experiments presented in Fig1.
2. Owing that the study implies effects of secreted components it would be better if authors would evaluate SASP and SASP regulatory pathways as an additional marker for senescence in addition to beta-gal, p16 and p21.
3. It is not clear what is the variability in the telomere FISH intensity between different repeats of the experiment as the graph 2c does not have any data that would allow this. I would suggest to find a way of presentation that allows to show the variability (especially if the legend says: data are means +SEM....)
4. The authors use very nice experimental system - liver explants - to show that neutrophils induce telomere dysfunction in hepatocytes. However, it is not clear what is allover effect of neutrophil addition to these explants. It is possible that neutrophils induce cell death (as might be pointed by increased LDH) and the observed effects are result of the death or are associated with it. It is important to evaluate the amount of cell death and that the TAFs are measured in live cells.
5. The conclusion that neutrophils affect senescence in acute liver damage is not supported by the presented data. I strongly suggest to reevaluate the results of this experiment and the conclusions drawn for several reasons delineated below. A. The markers authors used are not indicative for senescence on its own. Indeed p21 can be induced by many stimuli, including pro-apoptotic ones and it can be induced by toxins, like CCL4 in hepatocyte for the short time. The second marker the authors used is kariomegaly. The evidence that these hepatocytes are senescent are insufficient. The authors should provide co-stainings with additional markers for different aspects of senescent cells or reconsider their statement about senescence in this model. B. It is interesting to know if TAFs and indeed found more in hepatocytes with multiple copies of the genome, when corrected to the number of telomeres per genome. C. It is necessary to explain how senescence could be induced in these hepatocytes by CCL4 in very short period of time - 48h? D. it is necessary to know if 8-oxoDG positive cells and indeed TAF positive, kariamegaly, p21 positive. Without these co-stainings it is not possible to reach the conclusions authors suggest.

In general the conclusions that senescent hepatocytes are responsible for the recruitment of neutrophils to the liver is not supported by the presented data. The next few points should help to resolve the mysteries around this question.

6. The expression data of the livers with age is interesting, but it does not support by itself the conclusion that hepatocyte senescence is cause of increase in all the cytokines shown at Fig 5a. The cellular composition of the liver might change with age, and the increased amount of immune cells, which express higher levels of cytokines comparing to hepatocytes, could be the source of the observed changes in gene expression. Expression profile of hepatocytes from mice of different age could show if hepatocytes are indeed the source of cytokines that are able of recruiting the neutrophils.

7. The proximity of a hepatocyte and a neutrophil is not indicative of anything by itself. The quantitative analysis is necessary in order to support suggested conclusion. For example - how many neutrophils are in vicinity of TAF+ hepatocytes vs TAF- hepatocytes?
8. The in vivo data suggest that elimination of p16+ cells reduces the amount of neutrophils in the liver. However, there are variety of immune cells that express p16. These cells are eliminated by the AP treatment together with p16+ resident cells of the liver (hepatocytes, cholangiocytes...). FACS based quantification of different immune cell types in the liver and blood following the AP treatment would be necessary in order to start addressing this point. In addition, it is important to know if effect of hepatocytes in vivo is direct.
9. Injections of Cxcl1 are not sufficient to support the role of neutrophils or senescent cells. It is necessary to understand what are all other immune cells types that were recruited to the liver following the injections and what is their relative abundance and consequently their role in the process. The data as presented is correlative only and does not led to any conclusion relevant to the role of neutrophils. Specific elimination of the neutrophils in this system might help to establish a direct connection.

Referee #2:

In the manuscript by Lagnado et al., the authors propose a critical role for neutrophils inducing bystander senescence in normal naïve cells. The authors show in vitro, in co-culture experiments, that activated neutrophils induce a decrease in lifespan and the activation of senescence marker in human diploid fibroblasts. Besides, the authors show that exogenous infiltrating neutrophils in PCLS from human liver induce an acute senescence response in hepatocytes. In mouse, the authors show that depletion of neutrophils with neutralizing antibodies reduces the senescence load in a model of CCl₄ induced liver damage. Similar results were obtained using genetic models of neutrophil manipulation (Tlr2KO) and two additional models of chronic liver damage.

Conversely, the authors showed that the SASP has neutrophil chemoattractant properties in vitro, and senescent hepatocytes recruit neutrophils into the liver during ageing through the SASP. Genetic interventions to eliminate senescent hepatocytes (p16+) during ageing using the INK-ATTAC mouse model show that depletion of senescent cells reduced ageing dependent neutrophil infiltration in the liver. Moreover, administration of the recombinant SASP chemokine Cxcl1, or the DAMP LPS, induced neutrophil infiltration and the activation of senescent markers in mouse hepatocytes. Mechanistically, the authors propose that neutrophil induced bystander senescence is produced by ROS mediated telomere disfunction.

The role of neutrophils inducing bystander senescence is conceptually novel and could have a significant impact to fully understand the role of cellular senescence in ageing and age-related pathophysiological conditions, and the link between cellular senescence and inflammation. The experiments are in general well performed. However, the manuscript is too preliminary and lack of some critical controls which could lead to misinterpretation of the results. All those reasons prevent the recommendation for publishing the manuscript in its current form in The EMBO Journal.

Major points:

- 1- The authors propose that ROS mediated bystander senescence is mediated by telomere dysfunction: however, this is not substantiated by robust data, and other alternative mechanisms are possible. Activated neutrophils produce a proinflammatory secretome which is remarkably similar to the SASP. Neutrophils secrete factors such as TGF β , CXCL chemokines (IL-8, CXCL1 etc.), CCL chemokines (CCL2, CCL20), IL-1, IL6 etc., all of which have been involved in paracrine

senescence or the reinforcement of the senescent phenotype. Moreover, some of these factors could produce a ROS increase which contributes to the senescent phenotype. It is plausible that neutrophils transmit senescence by the contribution of these factors rather than ROS mediated. It is also possible that the addition of catalase to the co-culture experiments have an effect in reducing neutrophil proinflammatory activation, which may lead to reduced paracrine senescence response. To avoid this potential confounding effect of catalase on neutrophil activation, instead of co-cultures, experiments with conditioned media (CM) from neutrophils on target cells with the addition of catalase or other SASP inhibitors to dissect specific neutrophilic cytokine/chemokine effects (Receptor inhibitors or neutralizing antibodies) are better controlled. The analysis of the presence of known proinflammatory mediators with a role inducing paracrine senescence in neutrophil CM should also be conducted to inform about potential senescence mediators (e.g. ELISA or proteomics).

2- Other have previously shown that cell contact interaction contributes to secondary senescence via NOTCH signalling. Thus, a potential mechanism for the induction of bystander senescence is by contact interaction between neutrophils and the fibroblasts or hepatocytes in co-culture experiments. This possibility should be explored with CM or co-cultures with inserts.

3- There is a lack of consistency between the effect of neutrophils in vitro in co-cultures and in vivo. While the effects of neutrophils inducing fibroblasts bystander senescence are long term reduction of culture lifespan by telomere shortening, in human hepatocytes and liver damage experiments, there is an acute senescent-stress response with an absence of telomeric shortening. Why is this? Is this a consequence of inefficient experimental conditions due to sub-optimal culture conditions for primary neutrophils in DMEM or as a result of the short life of these cells in such conditions? Could experiments with enriched CM from neutrophils give a more robust and robust response? Could the use of cell lines such as NB4 or HL-60, which could differentiate to neutrophils to produce CM enriched in neutrophil-derived proinflammatory factors help to produce a more controlled and robust response?

4- The senescence markers used along the manuscript are correct in general. However, it lacks an analysis of SASP induction during the resulting bystander senescence response. The robustness and nature of the SASP could help to identify the nature of the senescence response.

5- Some early senescence markers could be detected before the proliferation effects are noticeable during replicative senescence. The authors should show the senescent markers at early time points at the end of the neutrophil treatment in figure 1. p15 has been shown to have an essential role in paracrine senescence as a TGF β response gene. Thus, the authors should show p15 induction in figure 1. Some other markers such as p21, p16 and the SASP by mRNA expression will improve the robustness of the data.

6- Results with fibroblasts expressing hTERT show some resistance to senescence induction by neutrophils in figure 2g. However, this data is not conclusive as it is not possible to know if at any given point in the future neutrophil treated cells will eventually senesce, by the contribution of the p16 pathway or any other telomere independent senescence pathway (e.g. SASP induced). Moreover, in figure 1, the authors show the activation of p16 during neutrophil induced senescence, which is not a direct target of telomere dysfunction. A better dissection of other senescence pathways by targeting the p53 and the p16/p15 tumour suppressor pathways should inform better of other potential upstream pathway contribution to neutrophil induced senescence.

7- Results from figure 3 are interesting, and the technology used is fascinating and promising for human tissue research. However, the statistical test in results in figure 3f and g showing induction of senescence is missing, or the results are not significant. Expression of additional markers of senescence such as SA- β Gal, tumour suppressor genes and SASP markers would improve the robustness of this data.

8- results in figure 4 are robust; however, some additional markers such as p16 and the SASP are missing.

9- TLR2 has been shown to have a critical role in regulating the SASP and the senescence response (Campisi, Hara, Acosta and Attanasio labs). In the experiments using Tlr2 KO mice, mice lacking Tlr2 may be resistant to CCl4 induced senescence rather than a specific role in neutrophil infiltration. To address the specific role of tlr2 mediated neutrophil recruitment, the authors would require a conditional Tlr2 knockout in the hematopoietic compartment.

10- In figure 5 authors shown elegantly that depletion of p16 positive cells in mature mice using the ink-attac model has an impact in neutrophil infiltration in the liver, and in p16 mRNA expression and the accumulation of TAF+ hepatocytes. However, p21 and p53 instead of p16 would be more informative here as a marker a DDR upon telomere shortening, which is the postulated mechanism through the manuscript.

11- Cxcl1 is a chemokine that signals through the Cxcr2 receptor, which has been involved in direct reinforcement of the senescence phenotype. Also, LPS signals through TLR4 receptor, which has been involved in SASP modulation by the Campisi group. Can the authors discard that the results shown in figures 5j-m are not the result of direct induction of senescence by SASP/DAMP factors instead of the neutrophil infiltration?

Other additional points:

Statistical test in figure 1h is missing.

In figure 2a, some statistical test has been performed. However, no information about the sample mean, standard error and the individual values is shown. These values should be provided.

Referee #3:

In the present article, the authors propose a model where neutrophils recruitment induces telomere dysfunction in a ROS-dependent manner, leading to hepatic neighboring cells senescence. Moreover, they show that neutrophil invasion increases with age and it is correlated with DNA damage activation at telomeres in hepatic cell. This neutrophil-induced paracrine senescence would be part of an intensifying process in which a few hepatic senescent cells recruit neutrophils to the liver exacerbating telomere dysfunction in the surrounding cells.

The concept present in this work is interesting, given that the paracrine induce-senescence between different cell types or organs at the systemic level is emerging in the aging field.

However, the data suffer from several uncertainties in the experimental settings and results that preclude any robust conclusion on telomere and in vivo role of neutrophils.

Major concerns :

A) One of the main overinterpretation is the role of telomeres, which is a central point in the paper. It is not clear at all from the presented data whether the neutrophil-induced senescence is triggered by ROS-induced telomere dysfunction for several reasons :

- the entire Figure 1 might be an artifact of high oxygen in vitro + neutrophils. This is further confirmed by the fact that there is 30% p16 positive cells in the starting population and then the number is not really going up much - are the cells already in a pre-senescent state? Also, 20% oxygen can drive DNA damage at telomeres, so the entire figure should be re-done in low oxygen conditions.

- The rescue with hTERT does not necessarily mean telomere function restoration, even if telomere length increased, since TERT can have extra-telomeric roles. Moreover, the authors have to show whether, in addition to TAF, the total non-telomeric DDR is also reduced upon hTERT ? if total DDR is also reduced, how to conclude that the TERT effect is a telomeric specific ?

- It is important to determine whether 8-oxo-dG accumulation is specific of telomeres or it is a general damage all over the genome (Figure 4m). This point is also important to support the hypothesis of neutrophil-induced specific telomere damage.

- Figures 2, 3 and 4 present telomere length data in presence or depletion of neutrophils. Fig 2f and 2g suggest that telomere shortening is required for neutrophil-induced premature growth arrest (replicative senescence). Nevertheless, any neutrophil-induced telomere shortening is observed in Fig 3h (PCLS) and 4n (mouse model : IgG or Ly6G). Are PCLS hTERT positive ? This difference in hTERT positive or negative cells should be addressed in the manuscript.

B) Fig 2c : How the authors explain that Neutrophils + LPS + Catalase condition preserves better telomere length compare to control and only catalase treatment ?

C) In the Figure 3, the information about healthy neutrophil donors is missing. It is relevant to know whether donors are age-matched with the PCLS patients.

D) LD activity 48 and 72h conditions are highly variable, in Figure 3d. Therefore, LD activity decrease over time cannot be concluded.

E) Overall, Figure 4 title and conclusion are not well substantiated by the experimental senescence data. Senescence argument is only supported by an increase of positive p21 and negative PCNA cells, however at 48h PCNA positive cells increases. In my opinion, at least other senescence marker is required to add the « senescence concept » to the conclusion of mice experiments presented here.

As well, in the Figure EV3 : Tlr2 -/- and NASH mice model results (number of TAFs) are not enough to conclude senescence induction but only about telomeric dysfunction. Authors misconceive telomere dysfunction and senescence.

F) The authors use anti Ly6G to deplete neutrophils (for instance fig 4). They should present a depletion control, at least in periphery showing the % of depletion of neutrophils.

G) For the chemotaxis experiment, it has been shown that cytokines in the microenvironment can regulate neutrophil migration and NETs production. Do the authors look whether NETs are implicated in the senescence induced neutrophil migration?

H) The nomenclature of the anti Ly6G antibody in the paper is confusing with the perfusion of neutrophil experiments. The authors, when they performed neutrophil depletion (figure 4a/4b for instance), said « As shown previously²⁵, livers exhibited increased neutrophil infiltrates (measured by NIMP1) at 48h, which were significantly reduced by Ly6G (Figure 4b). « This corresponds to the addition of an anti-Ly6G antibody and not to the addition of Ly6G. In the figure 4 legend they said « neutrophil neutralizing antibody Ly6G » which is not correct too. They should specify neutrophil neutralizing antibody against Ly6G. Moreover, the authors should specify directly the clone they used and not the commercial reference. After checking, they used the 1A8 clones. So either they said that they add 1A8 antibody or anti Ly6G antibody but not adding Ly6G that has no sense.

I) Activated neutrophils are short-lived so it is not clear how such a short exposure to whatever they produce can induce senescence in hepatocytes.

Minor concerns :

1. Fig 1.h : WB does not clearly reflect p21 expression upon catalase treatment.
2. The resolution of Figure 2e is not enough. In order to better distinguish presence or absence of colocalizations, a zoom is required for all the images of this figure.
3. In the Figure EV2a, the mean number of gH2AX increases in all neutrophils conditions, included +LPS+ catalase, nevertheless, the image selected as an example here does not show this gH2AX accumulation.
4. Second line of the 4th part : spelling mistake « woudn ».

We would like to thank the reviewers for their time and insightful comments which helped improve the quality of our study. We believe we have addressed most of the comments both through the inclusion of new experimental data, additional methodological detail and discussion. We have also made sure that we discuss more carefully the limitations of our data and do not overstate our conclusions.

Please see our point-by-point response to reviewers below.

Referee #1:

This study aims to understand the relationships between neutrophils and senescent cells. The authors suggest that neutrophils accelerate senescence *via* ROS-mediated telomere dysfunction. They show correlation between neutrophils and telomere dysfunction in acute liver damage. They also suggest that during ageing senescent cells recruit neutrophils to the liver and spread senescence to other cells *via* the recruited neutrophils.

This is a novel study that shows an interesting angle of the interaction of senescent cells with the immune system: recruitment of the neutrophils and subsequent possible senescence spread. The concepts described are very interesting. However, multiple aspects of the study, especially ones related to *in vivo* experiments are quite preliminary and do not allow reaching the conclusions suggested by the authors. It is necessary to provide more data in regard to presented *in vivo* experiments in order to get some minimal support to the interesting suggested conclusions.

Specific comments:

1. The authors suggest that the effect of neutrophils on proliferative lifespan is mediated by ROS. However in the data presented the effect of catalase, and in fact neutrophils, was only tested in the presence of LPS, which leads to overstimulation of the neutrophils and might affect MRC5 cells as well. Therefore, in order to support the conclusion suggested by the authors, it is necessary to test the effects of neutrophils without LPS in ALL the experiments presented in Fig 1.

Just as a clarification, we have incubated neutrophils for 1 hour with LPS, but then washed the neutrophils in order to remove any LPS present (thus it is unlikely that LPS would be affecting MRC5 fibroblasts directly). As the effect of unprimed neutrophils on senescence was relatively mild (see growth curve in Figure EV1c), we decided to do all the experiments involving co-culture of neutrophils and human fibroblasts using primed neutrophils.

In order to further reinforce our hypothesis that suggests a role for ROS in neutrophil-induced paracrine senescence, we have now added an additional experiment where we utilized mouse precision cut liver slices and exposed them to neutrophils isolated from mice with reduced ROS (due to expression of mitochondrial-targeted catalase)- (see revised Figure 3).

2. Owing that the study implies effects of secreted components it would be better if authors would evaluate SASP and SASP regulatory pathways as an additional marker for senescence in addition to beta-gal, p16 and p21.

We thank the reviewer for the suggestion. In order to characterize the SASP, we carried out a cytokine array in conditioned media (CM) collected from these cells after 8 and 28 days of culture. From the panel of 48 soluble proteins analyzed, we found that several factors were elevated in human fibroblasts that had been co-cultured with activated neutrophils and that catalase was able to prevent this effect (these data are included in Figure 1j and Figure EV1g). We have also measured the expression of major SASP factors IL-6 and IL-8 by qPCR and found them to be increased in fibroblasts that were co-cultured directly with neutrophils 20 days after co-culture (see Figure 2k,l).

3. It is not clear what is the variability in the telomere FISH intensity between different repeats of the experiment as the graph 2c does not have any data that would allow this. I would suggest to find a way of presentation that allows to show the variability (especially if the legend says: data are means +SEM....).

We now show the histograms for each individual telomere signal (Figure 2a) as well as the variability for each independent experiment (Figure 2c). We apologize for the mistake in the Figure legend.

4. The authors use very nice experimental system - liver explants - to show that neutrophils induce telomere dysfunction in hepatocytes. However, it is not clear what is all over effect of neutrophil addition to

these explants. It is possible that neutrophils induce cell death (as might be pointed by increased LDH) and the observed effects are result of the death or are associated with it. It is important to evaluate the amount of cell death and that the TAFs are measured in live cells.

We have repeated the lactate dehydrogenase (LD) activity measurements in all conditions and PCLS and found no significant differences. Additionally, we have performed TUNEL staining in the liver explants and found no significant differences with or without neutrophils.

5. The conclusion that neutrophils affect senescence in acute liver damage is not supported by the presented data. I strongly suggest reevaluating the results of this experiment and the conclusions drawn for several reasons delineated below.

A. The markers authors used are not indicative for senescence on its own. Indeed p21 can be induced by many stimuli, including pro-apoptotic ones and it can be induced by toxins, like CCL4 in hepatocyte for the short time. The second marker the authors used is karyomegaly. The evidence that these hepatocytes are senescent is insufficient. The authors should provide co-staining with additional markers for different aspects of senescent cells or reconsider their statement about senescence in this model.

As suggested by the reviewer, we have now extended considerably our evaluation of senescent markers: we have now measured p16 by RNA-ISH (revised Figure 5i), senescence-associated distension of satellites (SADS) by centromere FISH (revised Figure 5l) and LaminB1 (revised Figure EV4b,c). We have also evaluated expression of SASP components Cxcl1 (revised Figure 5j) and Il-1a (revised Figure EV4a) by RNA-ISH. We have also shown that hepatocytes which show karyomegaly are positive for p21 and TAF (revised Figure EV4d & e)

B. It is interesting to know if TAFs and indeed found more in hepatocytes with multiple copies of the genome, when corrected to the number of telomeres per genome.

This is an excellent point by the reviewer. However, TAF analysis was performed in 3µm thick sections- so it only captures a very small fraction of the entire nucleus of the hepatocyte- thus it would not be possible to obtain an accurate estimate of the total number of telomeres *per genome*.

C. It is necessary to explain how senescence could be induced in these hepatocytes by CCL4 in very short period of time - 48h?

The reviewer is correct that it is surprising that senescence can be induced in 48h (particularly given kinetic information we have mostly from human fibroblasts)- however, work by Dr. Stuart Forbes's lab showed similar time frames for induction of senescent markers with acute liver injuries with CCl₄ and acetaminophen (Bird *et al.* 2018 *Sci Transl Med*). At 2 days post injury they have observed the appearance of p21 positive hepatocytes (negative for BrdU), as well as the induction of other senescent markers. Recent data indicates that the kinetics of induction of senescence is very much cell-type and stress dependent: see for instance recent work that shows that acute injury in liver can induce a senescent like phenotype in a matter of hours (Chu *et al.* 2020 *Aging Cell*).

D. it is necessary to know if 8-oxoDG positive cells are indeed TAF positive, karyomegalic, p21 positive. Without these co-staining studies it is not possible to reach the conclusions authors suggest.

We have conducted Immuno-FISH to determine if 8-oxoDG co-localizes with telomeres and found that it increases 48h after CCl₄ and is reduced by anti-Ly6G (revised Figure 5m,n). We have also established that hepatocytes positive for TAF and p21 are karyomegalic. Other combinations of senescence-associated markers were not possible due to technical reasons related to the reactivity of antibodies.

In general the conclusion that senescent hepatocytes are responsible for the recruitment of neutrophils to the liver is not supported by the presented data. The next few points should help to resolve the mysteries around this question.

6. The expression data of the livers with age is interesting, but it does not support by itself the conclusion that hepatocyte senescence is cause of increase in all the cytokines shown at Fig 5a. The cellular composition of the liver might change with age, and the increased amount of immune cells, which express higher levels of cytokines comparing to hepatocytes, could be the source of the observed changes in

gene expression. Expression profile of hepatocytes from mice of different age could show if hepatocytes are indeed the source of cytokines that are able of recruiting the neutrophils.

We completely agree with the assessment from the reviewer. Our data does not allow us to determine what cell type is the source of the cytokines. Our purpose in showing these data was merely to point out that cytokines known to recruit neutrophils were generally increased during aging. While we cannot pinpoint the contribution of different cell-types in the liver which contribute to inflammation, there is a huge literature which supports that ER stress, telomere damage, epigenetic modifications in hepatocytes are major contributors and especially given that these cells make up the bulk of liver cell mass.

7. The proximity of a hepatocyte and a neutrophil is not indicative of anything by itself. The quantitative analysis is necessary in order to support suggested conclusion. For example - how many neutrophils are in vicinity of TAF⁺ hepatocytes vs. TAF⁻ hepatocytes?

We thank the reviewer for this excellent suggestion. We have performed 3D immune-FISH in old mice (29 month old) and quantified TAF in hepatocytes as a function of distance from Ly6G positive neutrophils. We observed in aged mice that hepatocytes located in close proximity to neutrophils contained higher TAF numbers and that TAF inversely correlated with distance from neutrophils (revised Figure 6d & e)

8. The *in vivo* data suggest that elimination of p16⁺ cells reduces the amount of neutrophils in the liver. However, there is a variety of immune cells that express p16. These cells are eliminated by the AP treatment together with p16⁺ resident cells of the liver (hepatocytes, cholangiocytes...). FACS based quantification of different immune cell types in the liver and blood following the AP treatment **would be necessary** in order to start addressing this point. In addition, it is important to know if effect of hepatocytes *in vivo* is direct.

We completely agree with the reviewer that other cell-types apart from hepatocytes express p16^{Ink4a} (we do not exclude that possibility and have made it clear in the manuscript). Besides, it has been shown that activated macrophages can express p16^{Ink4a} uncoupled from the induction of cellular senescence. Thus, it is possible that in the case of *INK-ATTAC* mice, treatment with AP kills cells based on high expression of p16^{Ink4a}, which may include activated immune cells such as macrophages.

As suggested by the reviewer, we isolated intrahepatic leukocytes from young and old mouse livers and analyzed them by cytometry by time of flight (CyTOF), which allows for mapping and discriminating between different immune cell types. We first found that p16^{Ink4a} was absent in neutrophils, which we also confirmed by analyzing single-cell RNA sequencing obtained from aged rats (Ma *et al.* 2020) and mice (Mogilenko *et al.* 2020). This suggests that the reduction of neutrophils is not due to clearance of p16^{Ink4a} positive neutrophils and is potentially the consequence of elimination of other cell-types. Our CyTOF analyses revealed that p16^{Ink4a} was expressed in monocyte-derived macrophages, macrophages, B cells and dendritic cells; however, the fraction of p16^{Ink4a} only increased significantly with age in dendritic cells. Thus, our data does not exclude the possibility that intrahepatic p16^{Ink4a} activated macrophages or other immune cells may contribute to some extent to the recruitment of neutrophils. We have made this clear in the discussion and also toned down some of our conclusions regarding a role for senescent cells as potential recruiters of neutrophils. For example, our current title for this section now reads: "p16^{Ink4a} positive cells recruit neutrophils during aging" and we made sure that the limitations of our data are clearly stated.

The reviewer also asked us to evaluate if p16^{Ink4a} positive immune cell-types were eliminated by AP in aged *INK-ATTAC* mice. We have performed this experiment as requested, however, due to unforeseen circumstances we unexpectedly lost 2 aged animals (the analysis was therefore insufficiently powered to reach definite conclusions). We hope that the reviewer understand that aging a new cohort of these animals will take a considerable amount of time and is in our view unlikely to alter significantly our conclusions.

9. Injections of Cxcl1 are not sufficient to support the role of neutrophils or senescent cells. It is necessary to understand what are all other immune cell types that were recruited to the liver following the injections and what is their relative abundance and consequently their role in the process. The data as presented are correlative only and do not lead to any conclusion relevant to the role of neutrophils. Specific elimination of the neutrophils in this system might help to establish a direct connection.

Cxcl1 is widely accepted in the field as being a powerful and predominantly neutrophilic chemoattractant in the mouse (see eg. Chang *et al.* Hepatology 2015), the only other cell type that might be recruited are Cxcr2+ monocytes, however in liver damage, monocytes are predominantly recruited through Ccl2/Ccr2. Having said that, while supportive of our hypothesis, we cannot exclude that other immune cells are equally recruited or importantly that Cxcl1 by itself leads to other effects. We have acknowledged the limitations of the data and highlighted the correlative nature of our data.

Referee #2:

In the manuscript by Lagnado, *et al.*, the authors propose a critical role for neutrophils inducing bystander senescence in normal naïve cells. The authors show *in vitro*, in co-culture experiments, that activated neutrophils induce a decrease in lifespan and the activation of senescence markers in human diploid fibroblasts. Besides, the authors show that exogenous infiltrating neutrophils in PCLS from human liver induce an acute senescence response in hepatocytes. In mice, the authors show that depletion of neutrophils with neutralizing antibodies reduces the senescence load in a model of CCl4 induced liver damage. Similar results were obtained using genetic models of neutrophil manipulation (Tlr2KO) and two additional models of chronic liver damage.

Conversely, the authors showed that the SASP has neutrophil chemoattractant properties *in vitro*, and senescent hepatocytes recruit neutrophils into the liver during ageing through the SASP. Genetic interventions to eliminate senescent hepatocytes (p16⁺) during ageing using the *INK-ATTAC* mouse model show that depletion of senescent cells reduced ageing-dependent neutrophil infiltration in the liver. Moreover, administration of the recombinant SASP chemokine Cxcl1, or the DAMP LPS, induced neutrophil infiltration and the activation of senescent markers in mouse hepatocytes. Mechanistically, the authors propose that neutrophil induced bystander senescence is produced by ROS mediated telomere dysfunction.

The role of neutrophils inducing bystander senescence is conceptually novel and could have a significant impact to fully understand the role of cellular senescence in ageing and age-related pathophysiological conditions, and the link between cellular senescence and inflammation. The experiments are in general well performed.

However, the manuscript is too preliminary and lacks some critical controls which could lead to misinterpretation of the results. All those reasons prevent the recommendation for publishing the manuscript in its current form in The EMBO Journal.

Major points:

1- The authors propose that ROS mediated bystander senescence is mediated by telomere dysfunction: however, this is not substantiated by robust data, and other alternative mechanisms are possible. Activated neutrophils produce a pro-inflammatory secretome which is remarkably similar to the SASP. Neutrophils secrete factors such as TGF β , CXCL chemokines (IL-8, CXCL1, *etc.*), CCL chemokines (CCL2, CCL20), IL-1, IL6, *etc.*, all of which have been involved in paracrine senescence or the reinforcement of the senescent phenotype. Moreover, some of these factors could produce a ROS increase, which contributes to the senescent phenotype. It is plausible that neutrophils transmit senescence by the contribution of these factors rather than ROS mediated. It is also possible that the addition of catalase to the co-culture experiments has an effect in reducing neutrophil pro-inflammatory activation, which may lead to reduced paracrine senescence response. To avoid this potential confounding effect of catalase on neutrophil activation, instead of co-cultures, experiments with conditioned media (CM) from neutrophils on target cells with the addition of catalase or other SASP inhibitors to dissect specific neutrophilic cytokine/chemokine effects (Receptor inhibitors or neutralizing antibodies) are better controlled. The analysis of the presence of known pro-inflammatory mediators with a role inducing paracrine senescence in neutrophil CM should also be conducted to inform about potential senescence mediators (*e.g.*, ELISA or proteomics).

We thank the reviewer for the helpful suggestion. In order to investigate these hypotheses, we co-cultured human fibroblasts with neutrophils for 3 days as described before and in parallel exposed human fibroblasts to conditioned media from neutrophils or co-cultured neutrophils and fibroblasts in separate

layers sharing the same medium using Transwell inserts (see revised Figure EV3a). We observed that human fibroblasts that had been in direct contact with neutrophils for 3 days experienced a premature growth arrest starting at 20 days of culture, while fibroblasts treated with conditioned medium or in transwells continued proliferation (see revised Figure EV3b). Furthermore, our results indicate that direct cell-to-cell contact between neutrophils and fibroblasts is necessary to induce TAF (see revised Figure EV3c-e), p21 expression (see revised Figure EV3f) and expression of SASP factors IL6 and IL-8 (see revised Figure EV3g & h).

2- Others have previously shown that cell contact interaction contributes to secondary senescence *via* NOTCH signaling. Thus, a potential mechanism for the induction of bystander senescence is by contact interaction between neutrophils and the fibroblasts or hepatocytes in co-culture experiments. This possibility should be explored with CM or co-cultures with inserts.

We agree. See response above.

3- There is a lack of consistency between the effect of neutrophils *in vitro* in co-cultures and *in vivo*. While the effects of neutrophils inducing fibroblasts' bystander senescence are long term reduction of culture lifespan by telomere shortening, in human hepatocytes and liver damage experiments, there is an acute senescent-stress response with an absence of telomeric shortening. Why is this? Is this a consequence of inefficient experimental conditions due to sub-optimal culture conditions for primary neutrophils in DMEM or as a result of the short life of these cells in such conditions? Could experiments with enriched CM from neutrophils give a more robust and robust response? Could the use of cell lines such as NB4 or HL-60, which could differentiate to neutrophils to produce CM enriched in neutrophil-derived pro-inflammatory factors help to produce a more controlled and robust response?

We agree that it is puzzling why mouse and human cells appear to respond differently. The lifespan of MRC5 is limited because of their susceptibility to replicative senescence, hence the very act of culturing is rendering them susceptible to telomere erosion and this is then exacerbated by exposure to neutrophils. By contrast, the hepatocytes in the PCLS and *in vivo* are predominantly quiescent and will only replicate in response to liver damage. As such, they will have less susceptibility to telomere erosion, but as previous data has shown, can accumulate stress-induced telomere damage irrespective of length. We have previously found that in liver with age, hepatocytes accumulate TAF, but do not show associated telomere shortening (Hewitt *et al.* 2012 *Nature Communications*). Fumagalli *et al.* 2012 *Nature Cell Biology*, showed that when mice are exposed to relatively mild stress (for instance sub-lethal doses of γ -irradiation), TAF are induced irrespective of telomere length and persist over long periods of time. We have made similar observations in other post-mitotic or quiescent cells such as cardiomyocytes (Anderson *et al.* 2019 *EMBO J*) or melanocytes (Vitorelli *et al.* 2019 *EMBO J*).

Additionally, in both *in vivo* (CCl₄ model) and *ex vivo* (PCLS) experiments we analyzed paracrine effects of neutrophils in hepatocytes after 48-72 hours and in order to observe significant telomere shortening (driven by cell division) longer periods of proliferation may be required.

With regards to the point about potential sub-optimal culture conditions for primary neutrophils, we have carefully monitored the viability of neutrophils for all our experiments and conducted the co-culture experiments under 3% Oxygen in order to improve their viability. We have also replenished neutrophils every 24h (all age-matched) so that they are not in culture for prolonged periods of time.

4- The senescence markers used along the manuscript are correct in general. However, it lacks an analysis of SASP induction during the resulting bystander senescence response. The robustness and nature of the SASP could help to identify the nature of the senescence response.

We agree with the suggestion. We have performed an additional analysis of SASP induction by ELISA and RT-PCR.

5- Some early senescence markers could be detected before the proliferation effects are noticeable during replicative senescence. The authors should show the senescent markers at early time points at the end of the neutrophil treatment in figure 1. p15 has been shown to have an essential role in paracrine senescence as a TGF β response gene. Thus, the authors should show p15 induction in figure 1. Some other markers such as p21, p16, and the SASP by mRNA expression will improve the robustness of the data.

We have evaluated p15 induction but did not find it to be reproducibly induced. We have also expanded the analysis of other senescent markers such as p21, p16 and SASP factors by mRNA (see revised Figure EV2)

6- Results with fibroblasts expressing hTERT show some resistance to senescence induction by neutrophils in figure 2g. However, these data are not conclusive as it is not possible to know if at any given point in the future neutrophil treated cells will eventually senesce, by the contribution of the p16 pathway or any other telomere independent senescence pathway (e.g., SASP-induced). Moreover, in figure 1, the authors show the activation of p16 during neutrophil-induced senescence, which is not a direct target of telomere dysfunction. A better dissection of other senescence pathways by targeting the p53 and the p16/p15 tumour suppressor pathways should inform better of other potential upstream pathway contribution to neutrophil induced senescence.

We have extended our characterization of hTERT expressing fibroblasts. Apart from no changes in telomere length or growth arrest, we observed that hTERT expressing fibroblasts did not show increased TAF, p21, p15 or p16 20 days after co-culture with neutrophils. We have now repeated this experiment independently 6 times and found no evidence for any long-term effects of neutrophil co-culture.

7- Results from figure 3 are interesting, and the technology used is fascinating and promising for human tissue research. However, the statistical test in results in figure 3f and g showing induction of senescence is missing, or the results are not significant. Expression of additional markers of senescence such as SA- β Gal, tumour suppressor genes, and SASP markers would improve the robustness of these data.

As requested, we have now extended the number of patients- a total of 7 human precision cut liver slices treated with or without neutrophils were analyzed. Data shows statistical significant increases in mean number of TAF and % of TAF positive hepatocytes. We have also conducted immunohistochemistry against senescence-associated markers p21 and p16^{INK4A} where we also observed significant increases upon addition of neutrophils.

8- results in figure 4 are robust; however, some additional markers such as p16 and the SASP are missing.

We now include RNA-ISH for p16^{Ink4a} and SASP factors (IL-1 α , Cxcl1). We have also evaluated senescence-associated distension of satellites (SADS) by centromere FISH (revised Figure 5l) and LaminB1 (revised Figure EV4b,c).

9- TLR2 has been shown to have a critical role in regulating the SASP and the senescence response (Campisi, Hara, Acosta, and Attanasio labs). In the experiments using Tlr2 KO mice, mice lacking Tlr2 may be resistant to CCl4 induced senescence rather than a specific role in neutrophil infiltration. To address the specific role of tlr2 mediated neutrophil recruitment, the authors would require a conditional Tlr2 knockout in the hematopoietic compartment.

We agree, data are correlative. We acknowledged this limitation in the results section and cite the aforementioned papers and a possible direct effect of Tlr2 in regulating the SASP.

10- In figure 5 authors shown elegantly that depletion of p16 positive cells in mature mice using the *INK-ATTAC* model has an impact in neutrophil infiltration in the liver, and in p16 mRNA expression and the accumulation of TAF⁺ hepatocytes. However, p21 and p53 instead of p16 would be more informative here as a marker a DDR upon telomere shortening, which is the postulated mechanism through the manuscript.

We now include now data showing p21 mRNA expression in INK-ATTAC mice following AP. Interestingly, we did not see any effect of AP on p21 mRNA expression (revised Extended Figure EV5c).

11- Cxcl1 is a chemokine that signals through the Cxcr2 receptor, which has been involved in direct reinforcement of the senescence phenotype. Also, LPS signals through TLR4 receptor, which has been involved in SASP modulation by the Campisi group. Can the authors discard that the results shown in figures 5j-m are not the result of direct induction of senescence by SASP/DAMP factors instead of the neutrophil infiltration?

We agree with the reviewer's comment. We have acknowledged this limitation in the discussion.

Other additional points:

Statistical test in figure 1h is missing.

We have done one-way ANOVA as in majority of multiple comparisons. However, in the case of p21, while all separate Western blots show exactly the same trend, when combined they are not statistically significant. As part of the revision, we have repeated the co-culture experiment and observed clear increases in p21 expression both at protein and mRNA level.

In figure 2a, some statistical test has been performed. However, no information about the sample mean, standard error, and the individual values is shown. These values should be provided.

We apologize and now provide this information. We performed Mann-Whitney test to investigate difference between distributions of telomere intensities, the blue dotted line represent median telomere FISH fluorescence intensity. We now show the mean \pm s.e.m of telomere FISH intensities from 3 independent experiments (see revised Figure 2c).

Referee #3:

In the present article, the authors propose a model where neutrophil recruitment induces telomere dysfunction in a ROS-dependent manner, leading to hepatic neighboring cells senescence. Moreover, they show that neutrophil invasion increases with age and it is correlated with DNA damage activation at telomeres in hepatic cell. This neutrophil-induced paracrine senescence would be part of an intensifying process in which a few hepatic senescent cells recruit neutrophils to the liver exacerbating telomere dysfunction in the surrounding cells.

The concept present in this work is interesting, given that the paracrine induced senescence between different cell types or organs at the systemic level is emerging in the aging field.

However, the data suffer from several uncertainties in the experimental settings and results that preclude any robust conclusion on telomere and *in vivo* role of neutrophils.

Major concerns:

A) One of the main over-interpretations is the role of telomeres, which is a central point in the paper. It is not clear at all from the presented data whether neutrophil-induced senescence is triggered by ROS-induced telomere dysfunction for several reasons:

- the entire Figure 1 might be an artifact of high oxygen *in vitro* + neutrophils. This is further confirmed by the fact that there are 30% p16 positive cells in the starting population and then the number is not really going up much - are the cells already in a pre-senescent state? Also, 20% oxygen can drive DNA damage at telomeres, so the entire figure should be re-done in low oxygen conditions.

We apologize for the lack of clarity- the co-culture *per se* was performed at 3% oxygen. Also, 30% of cells were not positive for p16 in the starting population. All the data presented in Figure 1 were analyzed 28 days following the 3 day co-culture, when the cells (both controls and neutrophil-treated cells) were close to transitioning into a senescent arrest.

We have now strengthen our hypothesis of a link between neutrophil-derived ROS and paracrine senescence by inclusion of a new set of data where we utilized mouse precision cut liver slices and exposed them to neutrophils with reduced ROS (derived from mice expressing mitochondrial targeted catalase)- (see revised Figure 3). We have observed that neutrophils isolated from the bone marrow of *MCAT* mice did not induce TAF or p21 in PCLS, which further supports that the involvement of neutrophil-derived ROS in the process. We have also conducted additional experiments now in revised Figure EV3 where we tested the role of longer-lived soluble factors released by neutrophils in the induction of paracrine senescence.

- The rescue with hTERT does not necessarily mean telomere function restoration, even if telomere length increased, since TERT can have extra-telomeric roles. Moreover, the authors have to show whether, in addition to TAF, the total non-telomeric DDR is also reduced upon hTERT? if total DDR is also reduced, how to conclude that the TERT effect is a telomeric specific?

We agree with the reviewer that hTERT can have non-canonical roles which are non-telomeric and this may impact on the cells resistance to neutrophil-induced stress. We have now carefully acknowledged the limitations of our data and its interpretation. However, as far as we know, there is no other mechanism we can explore experimentally to counteract telomere shortening. Ectopic expression of hTERT is established in the field as a way to determine telomere dependency (in fact, the reason we think that telomeres induce senescence is based on the work by Bodnar *et al.* 1998 showing that ectopic expression of hTERT bypasses senescence).

We have extended our characterization of senescent markers upon expression of hTERT (see revised Figure 2). We observed that expression of hTERT prevents neutrophil-induced telomere shortening and the induction of telomere-associated foci (revised Figure 2 g, h) as well as the induction of several senescence-associated markers- such as p21, p16 and SASP components (revised Figure 2 j-m). We have also observed that neutrophil co-culture did not elicit an increase in total DDR foci over time (see revised Figure EV2c). In terms of non-canonical roles of hTERT, this would be difficult to decipher experimentally. Previous work had shown that telomerase can shuttle from the nucleus to the mitochondria upon oxidative challenge, where it affects mitochondrial function and reduces ROS generation (Ahmed *et al.*, 2008). We did not observe any relocation of telomerase to the cytosol upon neutrophil co-culture (see revised Figure EV2), however, we cannot discard the possibility that telomerase may be conferring resistance to neutrophil-mediated paracrine damage in the nucleus via its non-canonical roles (this limitation is also discussed).

Our argument about the relative importance of telomere-associated damage is based on a body of work by our group and others: namely that damage at telomeric regions induces a permanent, unresolved DNA damage response (DDR) that is necessary to induce senescence (Hewitt, *et al.*, *Nature Comm* 2012). Telomeres (specifically telomere binding proteins- such as TRF2) inhibit NHEJ that prevents the resolution of the DDR (Fumagalli, *et al.*, *Nature Cell Biology* 2012). Additionally, we have shown that induction of double-stranded breaks to non-telomeric regions is insufficient to induce senescence (and can be easily repaired). However, when damage is induced specifically at telomeric regions, this leads to a persistent DDR and senescence is induced (Anderson, *et al.*, *EMBO J* 2019). Finally, we find that telomere damage cannot be dissociated from non-telomeric damage during senescence. We have shown that when telomeres become dysfunctional and activate the senescence program, they induce increased ROS production within 2-3 days, which leads to further DNA damage in a positive feedback loop (Passos, *et al.*, *Mol Sys Biol* 2010; Correia-Melo, *et al.*, *EMBO J* 2016). Overall, the data is conclusive in indicating that both TAF and non TAF exist in senescent cells, however, our interpretation is that TAF are the ones which elicit a persistent DDR capable of inducing and maintaining senescence.

- It is important to determine whether 8-oxo-dG accumulation is specific of telomeres or it is a general damage all over the genome (Figure 4m). This point is also important to support the hypothesis of neutrophil-induced specific telomere damage.

As requested, we have performed Immuno-FISH to determine co-localization between 8-oxodG and telomeres. We found that CCl₄ treatment led to an increase in 8-oxo-dG co-localization with telomeres at 48 hours (the same time point in which we observed increased TAF, p21, p16 and SADS) and this was reduced by anti-Ly6G.

- Figures 2, 3, and 4 present telomere length data in presence or depletion of neutrophils. Fig 2f and 2g suggest that telomere shortening is required for neutrophil-induced premature growth arrest (replicative senescence). Nevertheless, any neutrophil-induced telomere shortening is observed in Fig 3h (PCLS) and 4n (mouse model: IgG or Ly6G). Are PCLS hTERT positive? This difference in hTERT positive or negative cells should be addressed in the manuscript.

We agree with the reviewer that this is a puzzling observation: why do neutrophils elicit accelerated telomere shortening in human fibroblasts, but do not do so in liver hepatocytes from both mice and humans? One plausible explanation for this discrepancy we suspect may be related to the cells mitotic index. Human fibroblasts are proliferating cells and we observed that co-culture with neutrophils during 3

days does not affect their proliferation rate early on, but accelerates the onset of senescence at later stages. This is consistent with a model in which 3-day exposure to ROS-derived from neutrophils results in DNA damage and/or oxidative lesions at telomere regions that in combination with cell-division accelerates the rate of telomere shortening as demonstrated before (Petersen et al. 1998; von Zglinicki et al. 2020, Fouquerel et al. 2019). In fact, even if double-stranded breaks at telomeres are generated, it is possible that these can be repaired by homologous recombination in cells which are undergoing S-phase (Mao et al, 2016). Similarly, telomerase also requires transition into S-phase to play its role in telomere maintenance (Tomlinson *et al.* 2006).

In contrast, hepatocytes in tissues are mostly quiescent and only proliferate after induction of liver damage. For instance, Fumagalli and colleagues have shown that exposure of mice to low-doses of γ -irradiation, resulted in the induction of TAF in hepatocytes without telomere shortening and our group has reported that the age-dependent accumulation of TAF in hepatocytes from mice occurs irrespectively of telomere length (Hewitt et al. 2012 *Nature Communications*).

We have now included an additional paragraph in the discussion, addressing the discrepancies in telomere shortening in different models and organisms and possible explanations.

B) Fig 2c : How the authors explain that Neutrophils + LPS + Catalase condition preserves better telomere length compare to control and only catalase treatment?

It may seem that way (at day 4) based on the histogram; however, differences are not statistically significant.

C) In the Figure 3, the information about healthy neutrophil donors is missing. It is relevant to know whether donors are age-matched with the PCLS patients.

They are not age-matched. Liver donors were between 49-70 years of age and healthy neutrophils were isolated from healthy individuals aged between 21-30 years of age (with purity above 90%).

D) LD activity 48 and 72h conditions are highly variable, in Figure 3d. Therefore, LD activity decrease over time cannot be concluded.

We have repeated the lactate dehydrogenase (LD) activity measurements in all conditions and in PCLS from 7 patients and found no significant differences.

Additionally, we have performed TUNEL staining in the liver explants and found no significant differences with or without neutrophils suggesting that cell-death is not induced by neutrophils in PCLS.

E) Overall, Figure 4 title and conclusion are not well substantiated by the experimental senescence data. Senescence argument is only supported by an increase of positive p21 and negative PCNA cells, however at 48h PCNA positive cells increases. In my opinion, at least other senescence marker is required to add the « senescence concept » to the conclusion of mice experiments presented here.

As well, in the Figure EV3: Tlr2^{-/-} and NASH mice model results (number of TAFs) are not enough to conclude senescence induction but only about telomeric dysfunction. Authors misconceive telomere dysfunction and senescence.

We have now included several additional senescence-associated markers (see response to reviewers 1 & 2).

F) The authors use anti Ly6G to deplete neutrophils (for instance fig 4). They should present a depletion control, at least in periphery showing the % of depletion of neutrophils.

We have provided quantification of neutrophil infiltrations in the liver (NIMP⁺ve cells- in revised Figure 5b) and showed that anti-Ly6G significantly reduced neutrophil liver infiltrations. We have not examined frequency of neutrophils in the circulation in this specific experiment; however, have confirmed depletion in the circulation in separate experiments (with similar anti-Ly6G regimens).

G) For the chemotaxis experiment, it has been shown that cytokines in the microenvironment can regulate neutrophil migration and NETs production. Do the authors look whether NETs are implicated in the senescence induced neutrophil migration?

We have data (not shown) which indicates that exposure to senescence media induces NET formation in healthy neutrophils. We have not investigated if this is implicated in neutrophil migration. While an intriguing idea, we do not think it is necessary to support the main conclusions of this manuscript.

H) The nomenclature of the anti Ly6G antibody in the paper is confusing with the perfusion of neutrophil experiments. The authors, when they performed neutrophil depletion (figure 4a/4b for instance), said « As shown previously²⁵, livers exhibited increased neutrophil infiltrates (measured by NIMP1) at 48h, which were significantly reduced by Ly6G (Figure 4b). « This corresponds to the addition of an anti-Ly6G antibody and not to the addition of Ly6G. In the figure 4 legend they said « neutrophil neutralizing antibody Ly6G » which is not correct too. They should specify neutrophil neutralizing antibody against Ly6G. Moreover, the authors should specify directly the clone they used and not the commercial reference. After checking, they used the 1A8 clones. So either they said that they add 1A8 antibody or anti Ly6G antibody but not adding Ly6G that has no sense.

We thank the reviewer for the helpful suggestion. We have corrected it throughout the manuscript.

I) Activated neutrophils are short-lived so it is not clear how such a short exposure to whatever they produce can induce senescence in hepatocytes.

Neutrophils are important for resolution of inflammation and enabling the subsequent regenerative response. However, data is consistent in the literature suggesting that if damage is chronic, neutrophils fail to effectively resolve inflammation and can instead exacerbate cellular stress and injury. Our *in vitro*, *ex vivo* and *in vivo* data supports that a mere exposure to neutrophils for 3 days is sufficient to induce telomere dysfunction and premature senescence. We speculate that over time, neutrophil-induced senescence may be a contributor to tissue dysfunction during aging.

Minor concerns:

1. Fig 1.h: WB does not clearly reflect p21 expression upon catalase treatment.

All 3 independent western blots show exactly the same pattern and quantification is shown. We have now repeated the western blot in 3 separate co-culture experiments and confirmed p21 increase driven by neutrophil co-culture both at protein and mRNA level.

2. The resolution of Figure 2e is not enough. In order to better distinguish presence or absence of colocalizations, a zoom is required for all the images of this figure.

We provided better resolution images and indicate co-localizations.

3. In the Figure EV2a, the mean number of gH2AX increases in all neutrophils conditions, included +LPS+ catalase, nevertheless, the image selected as an example here does not show this gH2AX accumulation.

We now made sure foci are more clearly visible.

4. Second line of the 4th part: spelling mistake « woudn ».

Has been corrected.

Dear Joao,

Thank you for submitting your revised manuscript (EMBOJ-2020-106048R) to The EMBO Journal. Please accept my apologies for getting back to you with unusual protraction due to delayed reviewer input during re-review as well as detailed discussions here in the team. Your amended study was sent back to the three referees, and we have received comments from all of them, which I enclose below.

As you will see the referees stated that the manuscript has been significantly improved, and they are now broadly in favour of publication, pending satisfactory minor revision.

Thus, we are pleased to inform you that your manuscript has been accepted in principle for publication in The EMBO Journal.

Please consider the remaining points of the reviewers carefully by adding additional experimentation and re-analyses, or introducing caveats and more detailed discussion of the findings where appropriate.

Also, we need you to take care of a number of points related to formatting and data representation as detailed below, which should be addressed at re-submission.

Please contact me at any time if you have additional questions related to below points.

As you might remember from your previous work, every paper at the EMBO Journal now includes a 'Synopsis', displayed on the html and freely accessible to all readers. The synopsis includes a 'model' figure as well as 2-5 one-short-sentence bullet points that summarize the article. I would appreciate if you could provide this figure and the bullet points.

Thank you for giving us the chance to consider your manuscript for The EMBO Journal. I look forward to your adjusted manuscript files.

Again, we are happy to swiftly move forward with acceptance of this work upon re-submission. Please contact me at any time if you need any help or have further questions.

Kind regards,

Daniel

>> Introduce ORCID IDs for all corresponding authors (D.M.) via our online manuscript system.

Please see below for additional information.

>> Please specify distinct author contributions for S.E., D.G., K.P., H.R., S.P. and S.G. .

>> Recheck callouts and their correct order in the main text for Fig. 4I,J .

>> Please enter the complete funding information for your study into our online system: R01 AG050582 ; the Connor Fund ; the Noaber Foundation ; Robert and Theresa Ryan ; Ted Nash Long Life Foundation ; UL1 TR0002377 ; BB/L502066/1 ; W.E. harker Foundation ; Capes.

>> Dataset EV legends: Please add the legend to the respective excel table in a new table and correct the nomenclature to "Dataset EV1" and "Dataset EV2".

>>Please introduce a separate 'Data accessibility' section in the Material and Methods part, and indicate the access codes for the sequencing data on GEO or similar database as freely accessible entries. Please update the Author Checklist accordingly.

>> In line with the policies of our journal, we kindly ask you to provide uncropped source data for Fig. 2M.

>>Rename the 'Competing interests' section to 'Conflict of Interest'.

>> Please consider additional changes and comments from our production team as indicated by the .doc file enclosed and leave changes in track mode.

Please note that as of January 2016, our new EMBO Press policy asks for corresponding authors to link to their ORCID iDs. You can read about the change under "Authorship Guidelines" in the Guide to Authors here: <http://emboj.embopress.org/authorguide>

In order to link your ORCID iD to your account in our manuscript tracking system, please do the following:

1. Click the 'Modify Profile' link at the bottom of your homepage in our system.
2. On the next page you will see a box half-way down the page titled ORCID*. Below this box is red text reading 'To Register/Link to ORCID, click here'. Please follow that link: you will be taken to ORCID where you can log in to your account (or create an account if you don't have one)
3. You will then be asked to authorise Wiley to access your ORCID information. Once you have approved the linking, you will be brought back to our manuscript system.

We regret that we cannot do this linking on your behalf for security reasons. We also cannot add your ORCID iD number manually to our system because there is no way for us to authenticate this iD number with ORCID.

Thank you very much in advance.

Further information is available in our Guide For Authors:

The revision must be submitted online within 90 days; please click on the link below to submit the revision online before 26th Apr 2021.

Referee #1:

The revised version is a significant improvement of the original one. The authors did a great job on the revision.

Despite the progress this reviewer is still not convinced that "neutrophils drive hepatic senescence",

especially in the 48h time frame. I suggest being more careful about the definition and rather point on the result that pre-depletion of neutrophils affects the consequences of the damage induced by CCl₄ and expression of damage markers. It is not clear why to pursue the point that senescence can be induced during 48h in vivo while there is no molecular mechanism that can explain how they achieve stable arrest and SASP during this short period of time.

Referee #2:

The manuscript by Lagnado and collaborators have improved significantly. The data presented now is robust, showing a highly relevant role for neutrophils inducing cellular senescence. This data is fundamental for researchers in cellular senescence and its relation to ageing and inflammatory damage. The authors have addressed most of my concerns, showing that neutrophil effect is mediated by ROS and requires cell-cell contact interaction. In this regard, the experiments in mouse PCLS with neutrophils expressing MCAT are quite elegant and convincing. The statistical analysis of the data has also improved significantly. Thus, I recommend now the publication of the manuscript.

My only point for consideration would be the experiments with hTERT overexpression in human cells (in figure 2). In my opinion, these experiments do not add any mechanistic insight and only introduce confusion. The manuscript shows a clear effect of neutrophils in paracrine TAF induction independently of telomere length. However, I don't see any plausible explanation of why hTERT bypasses neutrophil induced TAF unless there is some unknown hTERT mechanism independent on telomere extension. My advice would be: or to remove this set of experiments from the manuscript (figures 2f-m), or to discuss in more detail this potential unknown mechanism (hTERT mediated repair, other?).

In any case, the manuscript should be published now.

Referee #3:

The answers and experiments provide by the authors in the revised version are still not fully convincing to confirm their previous interpretation that the ROS-mediated senescence induction is telomere dependent. As requested, the authors now give results showing that global DNA damage, including but not exclusively telomere damages, are induced by neutrophils showing that telomeres are a ROS targets among others. The fact that telomere damages are known to be irreparable does not imply that they are the only genomic regions being irreparable and that the telomere damages are those primarily triggering senescence. Along the same line, the authors underestimate the implication of the fact that TERT overexpression also rescued the total DNA damages, since this result precludes any firm conclusion on the genomic targets of the ROS. In fact, their results indicate that TERT overexpression has a global effect on genome stability. Thus, in order to really test the enrichment in telomere damages, this reviewer proposes to express their result not as TAF and global DDR separately but as a ratio of TAF by global DDR. In the context of a global DDR effect, this ratio will be crucial to investigate a telomere specific effect of their different settings.

From the two images shown for 8-oxo dG IF in the new figure 5 it is impossible to appreciate the specificity of the staining. More images must be shown in the supplementary files together with a counting of total 8-oxo-dG foci per nucleus.

Referee #1:

The revised version is a significant improvement of the original one. The authors did a great job on the revision.

We thank the reviewer for carefully evaluating our manuscript and the positive assessment of our revised work.

Despite the progress this reviewer is still not convinced that "neutrophils drive hepatic senescence", especially in the 48h time frame. I suggest being more careful about the definition and rather point on the result that pre-depletion of neutrophils affects the consequences of the damage induced by CCl₄ and expression of damage markers. It is not clear why to pursue the point that senescence can be induced during 48h in vivo while there is no molecular mechanism that can explain how they achieve stable arrest and SASP during this short period of time.

We understand and appreciate the concerns from the reviewer particularly since previously reported kinetics of senescence induction following acute damage, mostly in fibroblasts, show that weeks are required for the induction of the senescence phenotype and the SASP. We argue that it is plausible that the kinetics of senescence induction in hepatocytes is faster and this may be cell-type dependent, as shown in recent reports (Chu *et al.* 2020). However, we have changed the title and conclusions in this sub-section by replacing the term "senescence" with "senescence-associated pathways".

Referee #2:

The manuscript by Lagnado and collaborators have improved significantly. The data presented now is robust, showing a highly relevant role for neutrophils inducing cellular senescence. This data is fundamental for researchers in cellular senescence and its relation to ageing and inflammatory damage. The authors have addressed most of my concerns, showing that neutrophil effect is mediated by ROS and requires cell-cell contact interaction. In this regard, the experiments in mouse PCLS with neutrophils expressing MCAT are quite elegant and convincing. The statistical analysis of the data has also improved significantly. Thus, I recommend now the publication of the manuscript.

We thank the reviewer for the positive evaluation of our work and carefully evaluating our manuscript.

My only point for consideration would be the experiments with hTERT overexpression in human cells (in figure 2). In my opinion, these experiments do not add any mechanistic insight and only introduce confusion. The manuscript shows a clear effect of neutrophils in paracrine TAF induction independently of telomere length. However, I don't see any plausible explanation of why hTERT bypasses neutrophile induced TAF unless there is some unknown hTERT mechanism independent on telomere extension. My advice would be: or to remove this set of experiments from the manuscript (figures 2f-m), or to discuss in more detail this potential unknown mechanism (hTERT mediated repair, other?).

Our interpretation of the data is that in the case of proliferating cells such as fibroblasts, hTERT overexpression prevents neutrophil-induced telomere shortening and therefore the induction of TAF. We have however also observed that DNA damage foci present outside telomere regions are also significantly reduced, which could be suggestive of hTERT having non-canonical effects besides telomere elongation (eg. reducing DNA damage induction or improving overall DNA repair). We agree with the reviewer that this remains a possibility and impacts on the interpretation of the data.

There is however another possibility to explain the data. We have previously shown that upon telomere damage and activation of pathways downstream of the DDR, cells start producing high levels of ROS which contribute to secondary DNA damage (predominantly in non-telomeric regions) (Passos *et al.* 2010). Thus, it is possible that hTERT, by preventing telomere shortening and induction of TAF, is also reducing non-telomeric DNA damage induced by secondary ROS.

Either way, despite the caveats, we believe this is still a worthwhile experiment (particularly considering that there are not many other alternative ways to experimentally test telomere-dependency). As suggested by the reviewer, we have now extended our discussion and attempted to discuss more clearly the limitations of these experiments and suggest possible explanations.

Referee #3:

The answers and experiments provide by the authors in the revised version are still not fully convincing to confirm their previous interpretation that the ROS-mediated senescence induction is telomere dependent. As requested, the authors now give results showing that global DNA damage, including but not exclusively telomere damages, are induced by neutrophils showing that telomeres are a ROS targets among others. The fact that telomere damages are known to be irreparable does not imply that they are the only genomic regions being irreparable and that the telomere damages are those primarily triggering senescence. Along the same line, the authors underestimate the implication of the fact that TERT overexpression also rescued the total DNA damages, since this result precludes any firm conclusion on the genomic targets of the ROS. In fact, their results indicate that TERT overexpression has a global effect on genome stability. Thus, in order to really test the enrichment in telomere damages, this reviewer proposes to express their result not as TAF and global DDR separately but as a ratio of TAF by global DDR. In the context of a global DDR effect, this ratio will be crucial to investigate a telomere specific effect of their different settings.

We appreciate the well-reasoned arguments and consequently already in the previous version of the manuscript, limited any claims of telomere-dependency and carefully acknowledged possible limitations of our study. We should however emphasize that there is a considerable amount of literature from our laboratory and many others supporting that: i) telomere regions are highly susceptible to oxidative modifications; ii) telomere regions are more difficult to repair and inhibit Non-homologous end joining. Also, recent elegant work published by d'adda di Fagagna's laboratory has shown that inhibition of the DDR specifically at telomere regions (using sequence-specific telomeric antisense oligonucleotides) rescues cellular senescence

(Rossiello *et al.* 2017; Aguado *et al.* 2019 *Nature Comm*). This rescue occurs even though senescent cells contain a mixture of both non-telomeric and telomeric DDR and is suggestive of a key role for telomeric DDR in the induction and maintenance of senescence. Consistent with this, our group has shown that if we induce non telomeric DNA damage (using the homing endonuclease I-PpoI) most DNA damage is repaired within 4 days with no induction of senescent markers; however, if we induce similar numbers of DDR foci specifically at telomere regions (using a TRF1-FokI fusion protein) these are irreparable and we observe induction of senescent markers (Anderson *et al.* 2019 *EMBO J*).

For these reasons, we believe that presenting the data as a ratio between telomeric DNA damage foci and total DNA damage foci would not be a fair analysis of the data, since there is ample evidence that induction of senescence-pathways is highly dependent on the origin of the signal. We should also add that telomere regions occupy 0.01% of the entire genome, so it is more likely that stochastic oxidative damage would occur elsewhere in the genome. The fact that we observe a certain degree of oxidative damage at telomeres suggests that these regions may be inordinately susceptible to damage.

As initially suggested by the reviewer we have included information about both TAF and total DNA damage foci.

From the two images shown for 8oxo dG IF in the new figure 5 it is impossible to appreciate the specificity of the staining. More images must be shown in the supplementary files together with a counting of total 8-oxo-dG foci per nucleus.

As requested by the reviewer, we have now included additional images in appendix as well as a quantification of total 8-oxodG. Our data indicates that total 8-oxodG foci are not changed, only 8-oxodG co-localizing with telomeres.

Dear Joao,

Thank you for submitting the revised version of your manuscript. I have now evaluated your amended manuscript and concluded that the remaining minor concerns have been sufficiently addressed.

Thus, I am pleased to inform you that your manuscript has been accepted for publication in the EMBO Journal.

Please note that it is EMBO Journal policy for the transcript of the editorial process (containing referee reports and your response letter) to be published as an online supplement to each paper.

Also in case you might NOT want the transparent process file published at all, you will also need to inform us via email immediately. More information is available here:

http://emboj.embopress.org/about#Transparent_Process

Please note that in order to be able to start the production process, our publisher will need and contact you regarding the following forms:

- PAGE CHARGE AUTHORISATION (For Articles and Resources)

[http://onlinelibrary.wiley.com/journal/10.1002/\(ISSN\)1460-2075/homepage/tej_apc.pdf](http://onlinelibrary.wiley.com/journal/10.1002/(ISSN)1460-2075/homepage/tej_apc.pdf)

- LICENCE TO PUBLISH (for non-Open Access)

Your article cannot be published until the publisher has received the appropriate signed license agreement. Once your article has been received by Wiley for production you will receive an email from Wiley's Author Services system, which will ask you to log in and will present them with the appropriate license for completion.

- LICENCE TO PUBLISH for OPEN ACCESS papers

Authors of accepted peer-reviewed original research articles may choose to pay a fee in order for their published article to be made freely accessible to all online immediately upon publication. The EMBO Open fee is fixed at \$5,200 (+ VAT where applicable).

We offer two licenses for Open Access papers, CC-BY and CC-BY-NC-ND.

For more information on these licenses, please visit: <http://creativecommons.org/licenses/by/3.0/> and http://creativecommons.org/licenses/by-nc-nd/3.0/deed.en_US

- PAYMENT FOR OPEN ACCESS papers

You also need to complete our payment system for Open Access articles. Please follow this link and select EMBO Journal from the drop down list and then complete the payment process:

https://authorservices.wiley.com/bauthor/onlineopen_order.asp

Should you be planning a Press Release on your article, please get in contact with embojournal@wiley.com as early as possible, in order to coordinate publication and release dates.

On a different note, I would like to alert you that EMBO Press is currently developing a new format for a video-synopsis of work published with us, which essentially is a short, author-generated film explaining the core findings in hand drawings, and, as we believe, can be very useful to increase visibility of the work.

Please see the following link for a representative example:

The videos are embedded in the respective article html page, see e.g.

<https://www.embopress.org/doi/abs/10.15252/embojournal.2019103009>

If you have any questions, please do not hesitate to call or email the Editorial Office.

Kind regards,

Daniel

Daniel Klimmeck, PhD
Senior Editor
The EMBO Journal
EMBO
Postfach 1022-40
Meyerhofstrasse 1
D-69117 Heidelberg
contact@embojournal.org
Submit at: <http://emboj.msubmit.net>

Corresponding Author Name: Joao Passos

Journal Submitted to: EMBO J

Manuscript Number: EMBOJ-2020-106048